# COMPLEXITY-SEPARATED SCHEMES FOR ADDRESSING STRUCTURED HETEROGENEITY IN FEDERATED LEARNING

## ABSTRACT

Federated learning faces challenges due to heterogeneity in local training sets. Existing methods typically treat this as a monolithic challenge, leading to communication overhead. In this work, we suggest examining the structure of data heterogeneity in more detail. We identify two forms of this phenomenon: mode-based, where clients differ in the presence of common versus unique data modes; and coordinate-based, where groups of model parameters vary in statistical similarity. We develop algorithms that decouple communication complexity along these structural dimensions and consequently achieve reduced synchronization frequency without deterioration in convergence. Our analysis establishes the optimality of the proposed schemes. Extensive experiments on image and multimodal classification tasks demonstrate improvements in communication efficiency over state-of-the-art methods.

## 1 INTRODUCTION

Machine learning drives modern technological progress, from pattern recognition to complex predictive models (Shinde and Shah, 2018). In particular, advances in this field owe to the emergence of efficient optimization techniques that allow rapid adjustment of model parameters (Sun et al., 2019). Although initial successes were achieved in single-device settings, the scale of today's data have increasingly surpassed the limits of individual machines, prompting the need for distributed training (Verbraeken et al., 2020). It is typically organized in a server-worker architecture, where a powerful coordination hub (server) aggregates updates and maintains global model weights, and devices/clients/workers/nodes/machines perform local computations. In this paradigm, a shared dataset $\mathcal{D}$ is manually partitioned into $|M|$ disjoint subsets $\mathcal{D}_1, \ldots, \mathcal{D}_{|M|}$ distributed across machines. Each $m$-th one accesses only samples from $\mathcal{D}_m$ and calculates

$$h_m(x) = \frac{1}{|\mathcal{D}_m|} \sum_{(a,b) \in \mathcal{D}_m} \ell(u(x,a), b),$$

where $x$ is the parameters of the model $u$; $a$, $b$ are the vector representation and the label of the object from $\mathcal{D}_m$, respectively; and $\ell$ is the loss function. Minimizing the global objective is written down as

$$\min_{x \in \mathbb{R}^d} \left[ h(x) = \frac{1}{|M|} \sum_{m=1}^{|M|} h_m(x) \right]. \tag{1}$$

Distributed paradigm enables parallel computation to accelerate training. However, transmitting updates over the network becomes the primary constraint on learning speed, often exceeding computation time, particularly for large-scale models (Kairouz et al., 2021). The key performance metric of a numerical scheme is henceforth its communication efficiency in terms of the number of communication rounds (Kovalev et al., 2022), the total number of server-client vector exchanges (Lin et al., 2024), or the amount of transmitted bits (Beznosikov and Gasnikov, 2022), rather than the iterations count.

Various strategies have been developed to address the mentioned limitation (Seide et al., 2014; Alistarh et al., 2017; Stich, 2018). One of the possible ideas for overcoming the communication bottleneck is the use of statistical homogeneity. Since every $\mathcal{D}_m$ is the set of IID samples from the

global data distribution, each pair of devices has mutually aligned optimization landscapes. This phenomenon is commonly formalized via the bounded Hessian divergence condition (Shamir et al., 2014):

$$\|\nabla^2 h_i(x) - \nabla^2 h_j(x)\| \leq \delta_h, \ \forall x \in \mathbb{R}^d. \tag{2}$$

Smaller $\delta_h$ indicates higher data similarity, meaning local losses are consistent across the network. Crucially, it is known that $\delta_h$ typically decreases with growing data volume, as larger datasets better approximate the underlying distribution (Hendrikx et al., 2020). The stated property promotes the idea of utilizing local steps. Instead of synchronizing after every iteration, each worker performs an epoch of optimization and then transmits final parameters to the server to make the global update. Since $\delta_h$ in distributed networks is usually small, local gradients remain reliable estimators of global descent directions, ensuring communicative efficiency while preserving solution quality. To reduce computational overhead of devices, some papers use only the server to perform local steps, thus offloading devices (Hendrikx et al., 2020; Kovalev et al., 2022; Lin et al., 2024).

Despite the successes of the mentioned approach in distributed paradigm, some real-world scenarios pose challenges. In federated learning (FL), objects are generated by devices, while the server stores non-private data accumulated in public datasets (Konečný et al., 2016; Zhang et al., 2021). Therefore, the alignment of optimization landscapes is violated, and performing too many local updates steer the model toward inappropriate direction (Karimireddy et al., 2020, Table 3). To maintain convergence, algorithms must increase synchronization frequency, exacerbating communication bottleneck. However, there is an observation that helps to address this issue. A key idea underlying this work is that even heterogeneous networks exhibit structured patterns, manifesting in two distinct ways: how distribution modes are shared across clients, and how diverse are components of model parameters.

**Mixed heterogeneity in distribution patterns.** Focusing on distribution patterns, we find out that the training set can be divided into two parts. First one consists of ordinary objects *similar* to those contained in public datasets. Server-side and average losses computed on samples related to such modes express a high degree of statistical similarity. The second part is made up of unique data, that is *poorly represented* by the server storing little or none of corresponding modes. A vivid example is training a federated medical diagnostic model. The server may possess a large amount of scans showing widespread diseases, such as pneumonia or fractures, and many hospitals also have data on these pathologies. However, a small amount of specialized clinics may additionally store unique images of rare genetic syndromes, which are absent from the server. Similar structures arise in other federated learning domains (Kairouz et al., 2021). A natural mathematical model to describe such scenarios is a composite minimization problem

$$\min_{x \in \mathbb{R}^d} \left[ h(x) = f(x) + g(x) \right],$$

$$\text{with } f(x) = \frac{1}{|M_f|} \sum_{m \in M_f} f_m(x),$$

$$g(x) = \frac{1}{|M_g|} \sum_{m \in M_g} g_m(x), \tag{3}$$

where $f_m$, $g_m$ are the local losses calculated over data from frequent and rare modes, respectively; $M_f$, $M_g$ are the sets of clients containing non-zero $f_m$, $g_m$, respectively; $|M_f|$, $|M_g|$ are the cardinalities of these sets. We point out that $|M_g| \ll |M_f|$ in many practical applications (Li et al., 2022, Section 4). Thus, it is the interaction with $M_f$ that creates the bottleneck. Consequently, there is a potential for gain by communicating with nodes from $M_f$ and $M_g$ at different frequencies.

**Mixed heterogeneity in model parameters.** Returning to the example of medical federated learning, suppose that patient's metadata (for example, blood tests) is also available. For such tasks, a common approach is to train a network responsible for feature extraction from images, then concatenating its output with tabular data, and feeding the combined representation into a shared layer (Gao et al., 2020). Formally, $h_m$ from equation 1 takes the form

$$h_m(x, y) = \frac{1}{|\mathcal{D}_m|} \sum_{(a_1, a_2, b) \in \mathcal{D}_m} \ell(u(F_m(a_1, x), a_2, y), b),$$

where the object consists of two modalities $a_1$, $a_2$; $F_m$, $u$ are the encoder and the head, respectively; $x$, $y$ are the weights of corresponding models. It is established in literature that images have considerably more homogeneous embeddings than tabular inputs (Liang et al., 2022; Rabbani et al., 2024). This fact creates potential for less frequent updates of statistics within $x$ than within $y$. Additionally, processing structured metadata typically involves far fewer parameters compared to those used for extracting features from scans. Consequently, updating $x$ is significantly more expensive in terms of communicated information. Expanding this observation, we obtain the second formulation of interest:

$$\min_{(x,y)\in\mathbb{R}^{d_x}\times\mathbb{R}^{d_y}} \left[ h(x,y) = \frac{1}{|M|} \sum_{m=1}^{|M|} h_m(x,y) \right], \tag{4}$$

where the second derivatives of local objectives within $x$ exhibit more statistical similarity with the server than the ones within $y$. This setting promises efficiency gains through asymmetric update frequencies. In the case where $d_x \gg d_y$, the proposed setting offers potential to reduce the amount of information transferred from devices to the server in comparison to existing techniques that ignore this feature. The high-dimensional $x$ tolerate infrequent synchronization due to stable Hessian characteristics, while compact $y$ require regular but lightweight exchanges.

## 2 NOTATION

When analyzing the communication efficiency of federated learning schemes, it is important to choose an appropriate complexity measure. In this paper, we use three definitions emerging in literature to reach the full potential of proposed approaches.

• **Number of communication rounds.** In several works, the complexity of federated learning algorithms is analyzed without reference to the number of machines involved in each round of communication (Shamir et al., 2014; Kovalev et al., 2022). This metric suits for synchronous networks, where it only matters how many times the server accesses clients data during the training process.

• **Number of client-server communications.** For asynchronous networks, the number of rounds is inadequate. In such case, each server-client vector exchange should be counted as a complexity unit. This definition is well-established in the optimization community (Khaled and Jin, 2022; Lin et al., 2024). In our paper, we utilize it to analyze the distribution-based structured heterogeneity.

• **Number of communicated coordinates.** In addition to mentioned approaches, it is also common to analyze the complexity in terms of the number of communicated coordinates. Originally, this metric was designed for emphasizing the advantage of methods that reduce the size of transmitted vectors, e.g. for schemes with compression (Beznosikov and Gasnikov, 2022). In our work, we use it to derive results in the case of structured heterogeneity in model parameters.

Our study assumes the presence of independently accessible oracles either for aggregating over a group of nodes or for computing statistics within a block of parameters. Since the main goal of this paper is to obtain theoretical guarantees of acceleration for complex-structured objectives, the notion of complexity is applied to each of them individually.

## 3 OUR CONTRIBUTION

While existing federated learning methods treat heterogeneity as a monolithic challenge requiring uniform communication strategies, we develop techniques that decouple optimization complexity by accounting the structure of the objective in greater detail. We specifically focus on the non-convex setting, which remains under-explored in the works on data similarity despite its critical importance for modern applications. We formulate the list of our contribution as follows:

• **First method for distribution-based heterogeneity.** For the non-convex problem 3, we design a method that theoretically dominates existing techniques both in terms of communication rounds and client-server vector exchanges.

• **First method for coordinate-based heterogeneity.** For the non-convex problem 4, we propose a scheme that theoretically outperforms state-of-the-art heterogeneity-accounted techniques in the sense of communicated coordinates.

• **Optimality.** For the non-convex problem 3 with separate oracles $\nabla f$, $\nabla g$, we show the optimality of our method in terms of synchronizations count.

• **Empirical validation.** To support the theory, we compare our approach to modern methods for combating heterogeneity and state-of-the-art optimizer `Adam`. Numerical experiments include classification on *CIFAR-10* with *ResNet-18*, and searching for duplicate ads in *Avito* multimodal dataset with *BERT* and *ResNet-18*. The results show promising advantage over chosen baselines.

## 4 RELATED WORKS

### 4.1 COMPLEXITY SEPARATION

Classic works on numerical methods considered a single-machine minimization without assuming any additional structure of the objective (Polyak, 1987). For now, there are a lot of works devoted to the problem of oracle complexity separation when minimizing complex-structured functions. Below we provide a detailed review on this issue.

**Composite-sum case.** This setup considers $h(x) = f(x) + g(x)$ as the objective with separate oracles accessible for the components. Initial research in this direction was motivated by machine learning applications, where the empirical loss $f$ is usually regularized by a non-smooth function $g$ with easily computable statistics to avoid unbounded growth of model parameters. Thus, basic schemes are designed to handle the case where any optimization problem on $g$ has negligible complexity (Parikh et al., 2014). However, in many practical tasks, the mentioned property is not satisfied (Colson et al., 2007), and calls of the oracle corresponding to $g$ must not be unboundedly frequent, as in naive proximal schemes. To address this issue, Juditsky et al. (2011) applied an extragradient type algorithm to variational inequality reformulation of the initial problem and derived $\mathcal{O}\left(L/\varepsilon + M^2/\varepsilon^2\right)$ of both oracles calls in the convex case. Here $L$, $M$ are the Lipschitz constants of $\nabla f$, $g$, respectively, and $\varepsilon$ is the accuracy of the numerical solution. This result was enhanced to $\mathcal{O}\left(\sqrt{L/\varepsilon} + M^2/\varepsilon^2\right)$ by utilizing Nesterov's acceleration in (Lan, 2012). However, this rate is optimal only if oracles associated with $f$ and $g$ are not accessible separately. Assuming that the relevant statistics can be computed independently of each other, Lan (2016) obtained $\mathcal{O}\left(\sqrt{L_f/\varepsilon}\right)$ for $\nabla f$ and $\mathcal{O}\left(\sqrt{L_f/\varepsilon} + L_g^2/\varepsilon^2\right)$ for $g' \in \partial g$. The proposed `Gradient Sliding` guarantees that number of $\nabla f$ evaluations does not depend on the optimization landscape of $g$. To the best of our knowledge, it is the first algorithm that has progressed to split oracle complexities. The exact separation was also derived for *smooth+smooth* problems by Lan and Ouyang (2016). Their method achieves $\mathcal{O}\left(\sqrt{L_f/\varepsilon}\right)$, $\mathcal{O}\left(\sqrt{L_g/\varepsilon}\right)$ for convex objectives and $\tilde{\mathcal{O}}\left(\sqrt{L_f/\mu}\right)$, $\tilde{\mathcal{O}}\left(\sqrt{L_g/\mu}\right)$ for $\mu$-strongly convex ones.

At the moment, complexity separation is an established area of optimization. There are many exotic sliding-based schemes: for VIs (Lan and Ouyang, 2021; Emelyanov et al., 2024), saddle-points (Lan and Zhou, 2018; Chen et al., 2020; Tominin et al., 2021; Kuruzov et al., 2022; Borodich et al., 2023; Kovalev and Borodich, 2024), zero-order optimization problems (Beznosikov et al., 2020; Stepanov et al., 2021; Ivanova et al., 2022), high-order minimization (Kamzolov et al., 2020; Gasnikov et al., 2021; Grapiglia and Nesterov, 2023).

**Block-coordinate case.** Block-coordinate methods were also originally studied for minimizing $h(x, y)$ in isolation from the federated setting (Nesterov, 2012; Richtárik and Takáč, 2014; Nesterov and Stich, 2017). For small-scale problems, Gladin et al. (2021a) obtained $\tilde{\mathcal{O}}\left((d_x + d_y)\right)$, $\tilde{\mathcal{O}}\left((d_x + d_y)\sqrt{(L_x+L_y)/\mu}\right)$ of $\nabla_x h$, $\nabla_y h$ calls, respectively. Here $d_x$, $d_y$ are the dimensionalities of $x$, $y$; $L_x$, $L_y$ are the smoothness constants of $h$ in $x$, $y$; $\mu$ is the strong convexity constant. The first step to separation was made in (Gladin et al., 2021b). The complexities were $\tilde{\mathcal{O}}\left((d_x + d_y)\right)$ for $\nabla_x h$ and $\tilde{\mathcal{O}}\left(d_x d_y \sqrt{L_y/\mu}\right)$ for $\nabla_y h$. However, in the large-scale case this approach gives $\tilde{\mathcal{O}}\left(\sqrt{L_x/\mu}\right)$ and $\tilde{\mathcal{O}}\left(\sqrt{L_x L_y/\mu^2}\right)$, which is much worse than a desirable result for evaluations of $\nabla_y h$. This issues was addressed in (Gasnikov et al., 2022), where the `BAM` algorithm achieved $\tilde{\mathcal{O}}\left(\sqrt{L_x/\mu}\right)$ and $\tilde{\mathcal{O}}\left(\sqrt{L_y/\mu}\right)$.

## 4.2 HESSIAN SIMILARITY

Federated approaches that exploit data similarity rely on a simple but crucial trick. The objective $h$ defined in equation 1 is artificially rewritten as $h(x) = h_1(x) + (h - h_1)(x)$. Here $h_1$ belongs to the server and therefore computation of its statistics does not require exchanging information, and $(h - h_1)$ is related to clients. The idea of saving iterations by using a proximal friendly regularizer can be transferred to the federated setting to communicate less by utilizing local steps on the server. The main challenge in this direction is that theory for handling composite structure were initially developed for *convex+convex* case, while this setting is *convex+non-convex*.

The first approach addressing similarity was the Newton-type method, DANE, achieving $\tilde{\mathcal{O}}\left(\delta_h^2/\mu^2\right)$ communication rounds for quadratic $\mu$-strongly convex objectives (Shamir et al., 2014). For the class of problems under consideration, Arjevani and Shamir (2015) established a lower bound $\tilde{\Omega}\left(\sqrt{\delta_h/\mu}\right)$. This prompted the question of how to bridge the gap in complexities. Numerous papers explored this issue but either fell short of meeting the exact bound or required specific cases and unnatural assumptions (Zhang and Lin, 2015; Lu et al., 2018; Yuan and Li, 2020; Beznosikov et al., 2021; Tian et al., 2022). Recently, optimal rate in terms of rounds count was achieved by Kovalev et al. (2022). Most numerical schemes for the data similarity scenario were developed under fairly strong assumptions of the (strong) convexity of the objective. Non-convex problems were investigated in (Woodworth et al., 2023) that, however, failed to establish convergence to an arbitrary $\varepsilon$-solution.

## 5 SETUP

Machine learning applications, particularly deep neural networks, operate in fundamentally non-convex scenario (Cybenko, 1989; Nguyen and Hein, 2018). To address the contemporary challenges, we keep our theoretical restrictions minimal. Throughout this work, we rely on the following mild assumption.

**Assumption 1.** *The function $h : \mathbb{R}^d \to \mathbb{R}$ attains its minimum, i.e. there exists such $x^* \in \mathbb{R}^d$ that*
$$h(x^*) = \inf_{x \in \mathbb{R}^d} h(x) > -\infty.$$

This requirement is satisfied by most practical loss functions and is widely used in literature (Malitsky and Mishchenko, 2019; Li et al., 2021; Zhao et al., 2021). Further, based on the standard notion of data similarity, we formalize the intuition from Section 1 by quantifying structured heterogeneity through the gap between the server-side and the mean Hessians.

**Assumption 2.** *The functions $h, h_1$ in the problem 3 are $(\delta_f, \delta_g)$-related, i.e. for every $x \in \mathbb{R}^d$*
$$\|\nabla^2 f_1(x) - \nabla^2 f(x)\| \le \delta_f, \quad \|\nabla^2 g_1(x) - \nabla^2 g(x)\| \le \delta_g.$$

**Assumption 3.** *The functions $h, h_1$ in equation 4 are $(\delta_x, \delta_y, \delta_{xy})$-related, i.e. for every $(x_1, x_2) \in \mathbb{R}^{d_1} \times \mathbb{R}^{d_2}$*
$$\|\nabla^2_{xx} h_1(x, y) - \nabla^2_{xx} h(x, y)\| \le \delta_x, \quad \|\nabla^2_{yy} h_1(x, y) - \nabla^2_{yy} h(x, y)\| \le \delta_y,$$
$$\|\nabla^2_{xy} h_1(x, y) - \nabla^2_{xy} h(x, y)\| \le \delta_{xy}.$$

Without loss of generality, we consider $\delta_f \le \delta_g$ and $\delta_x \le \delta_y$. In the case where there is no shift in modes distribution or in coincidence of loss landscapes in groups of parameters, Assumptions 2, 3 are equivalent to the standard bounded heterogeneity. This does not narrow the generality with respect to works that deal with $L$-smooth objectives, since $\delta_h \sim {}^L/_{|\mathcal{D}_1|}$ (Hendrikx et al., 2020).

## 6 ALGORITHMS AND ANALYSIS

### 6.1 MODE-BASED STRUCTURED HETEROGENEITY

To develop the idea proposed in Section 1, we present **H**eterogeneity-**A**ware **S**kipped **C**lient **A**ggregation (HASCA for the non-convex problem 3. Algorithm 1 can be viewed as a natural development of Proximal Descent (Hendrikx et al., 2020; Woodworth et al., 2023), which underlies most state-of-the-art schemes leveraging data similarity in the (strongly) convex case. As discussed in Section 4, the key idea behind this approach is to artificially rearrange the objective as $h(x) = (h - h_1)(x) + h_1(x)$. Here, $h_1$ serves as a proximal friendly regularizer in the sense that

any optimization problem involving $h_1$ can be solved without interaction with the devices. Thus, the server can perform several steps of local optimization after each interaction with the clients, significantly reducing the bottleneck. This yields the update based on the minimization of

$$\tilde{A}_\theta^t(x) = \langle \nabla(h - h_1)(x^t), x \rangle + \frac{1}{2\theta}\|x - x^t\|^2 + h_1(x),$$

where forming the surrogate $\tilde{A}_\theta^t$ and solving the resulting subproblem happen entirely on the server. Our method is built upon the same intuition (see Line 3). However, the goal of our work is to construct a first-order scheme that accounts for $\delta_f < \delta_g$. To satisfy this requirement, Algorithm 1 reuses the most recent values of $\nabla f$, while $\nabla g$ is called at each iteration (Line 2). We emphasize that the use of $h_1$ in Line 3 does not increase the communication complexity within $M_f$, since the statistics of this function are computed on the server. To balance the quality of approximation with the cost of expensive synchronization, we introduce a reference point $w^t$ that is refreshed with some probability $p$ (Line 4). When $\delta_f/\delta_g = 1$, Algorithm 1 should reduce to standard `Proximal Descent`, i.e. $p = 1$. As this ratio decreases, the probability $p$ should decrease accordingly.

---

**Algorithm 1** HASCA

---

**Input:** initial points $x^0, w^0 \in \mathbb{R}^d$, number of iterations $T$
**Hyperparameters:** step size $\theta > 0$, probability of full aggregation $p \in (0, 1)$
1: **for** $t = 0, 1, \ldots, T - 1$ **do**
2:     $e^t = \nabla(f - f_1)(w^t) + \nabla(g - g_1)(x^t)$
3:     $x^{t+1} = \arg\min_{x \in \mathbb{R}^d} [A_\theta^t(x)]$, where

$$A_\theta^t(x) = \langle e^t, x \rangle + \frac{1}{2\theta}\|x - x^t\|^2 + h_1(x)$$

4:     $\omega^{t+1} = \begin{cases} x^{t+1} & \text{with probability } p \\ \omega^t & \text{with probability } 1 - p \end{cases}$

5: **end for**
6: **Output:** $x^T$

---

The update of $e^t$ (see Line 2 of Algorithm 1) enables an asymmetric interaction with $M_f$ and $M_g$, but also introduces obstacles that prevent a direct adaptation of known stochastic schemes to our setting. Before proceeding to the main results, we propose the following bound.

**Lemma 1.** *Suppose Assumptions 1, 2 hold. Then, for Algorithm 1 it implies*

$$\mathbb{E}_{e^{t+1}}\left[\|e^{t+1} - \nabla(h - h_1)(x^{t+1})\|^2\right] \leq \left(1 - \frac{p}{2}\right)\|e^t - \nabla(h - h_1)(x^t)\|^2 + \frac{2}{p}\delta_f^2\|x^{t+1} - x^t\|^2.$$

In Lemma 1, the deteriorating factor $1/p$ can be compensated by the relative smallness of $\delta_f$ compared to $\delta_g$. Designing an appropriate update rule for $e^t$ that yields the recurrence of this form is one of the main theoretical challenges of this work. Indeed, if $e^t$ is chosen improperly, the second term of the inequality becomes too large to ensure the desired convergence rate. Now that the intuition behind Algorithm 1 is clear, we move on to its iterative complexity.

**Theorem 1.** *Suppose Assumptions 1, 2 hold. Consider $\theta \leq \min\{1/8(\delta_f + \delta_g), p/8\sqrt{2}\delta_f\}$. Then, Algorithm 1 requires*

$$\mathcal{O}\left(\frac{\delta_f + \delta_g}{\varepsilon^2} + \frac{\delta_f}{p\varepsilon^2}\right) \text{ iterations}$$

*to achieve an arbitrary $\varepsilon$-solution, where $\varepsilon^2 = \mathbb{E}\left[\left\|\frac{1}{T}\sum_{t=1}^T \nabla h(x^t)\right\|^2\right]$.*

In particular, this result shows that if the update of $e^t$ were implemented with three options, including the separate call of $\nabla f$, achieving a comparable result would not be feasible. Since $\nabla f$ is communicated with probability $p$, it is possible to provide a corollary of Theorem 1.

**Corollary 1.** *Consider the conditions of Theorem 1. Algorithm 1 with $p = \delta_f/(\delta_f + \delta_g)$ requires*

$$\mathcal{O}\left(\frac{\delta_f}{\varepsilon^2}\right), \ \mathcal{O}\left(\frac{\delta_g}{\varepsilon^2}\right) \text{ calls of } \nabla f, \ \nabla g$$

*to reach an arbitrary $\varepsilon$-solution.*

Algorithm 1 outperforms existing approaches. Its closest competitor, `ProxyProx` (Woodworth et al., 2023), requires $\mathcal{O}\left((\delta_f+\delta_g)/\varepsilon^2\right)$ calls of both oracles. Thus, in this method, $\nabla f$ is communicated $\mathcal{O}\left(\delta_g/\delta_f\right)$ times more frequent than necessary under the structured heterogeneity regime. This overhead may be significantly large in practice, as some modes of distribution are poorly represented (or entirely absent) on the server side due to the local nature of data sources, leading to $\delta_g/\delta_f \gg 1$. Moreover, the improvement in terms of server–client exchanges may also be substantial, amounting to $\mathcal{O}\left((1 + |M_g|/|M_f|)\,\delta_g/\delta_f\right)$ times.

### 6.1.1 LOWER BOUNDS

To obtain upper bounds, the convergence of a specific method is derived for an arbitrary function without strengthening the assumptions. In contrast, establishing lower bounds is more challenging, as it requires constructing a specific example on which any algorithm from the considered class cannot perform better than a certain complexity threshold. To specify the schemes under consideration, we utilize the Proximal Incremental First-Order Oracle (Woodworth and Srebro, 2016), which is defined as $r_{f_1}^P(x,\theta) = [h_1(x), \nabla h_1(x), \mathrm{prox}_{\theta h_1}(x)]$ with $\theta > 0$. Assuming that the server has access to $r_{h_1}^P$, we determine the following class of algorithms.

**Definition 1.** *Consider a randomized algorithm $\mathcal{A}$ to solve the problem 3. In a synchronization round $t$, the server has two options. It can communicate all the clients and aggregate $\nabla(h-h_1)(x^t)$, or interact with devices from $M_g$ only and compute $\nabla(g - g_1)(x^t)$. Afterwards, it updates the information set based on the linear span operation and its local oracle $r_{h_1}^P$.*

Our analysis of lower bounds is based on techniques typically utilized for non-convex (Carmon et al., 2017) and homogeneous (Arjevani and Shamir, 2015) scenarios. To construct the hard instance of the problem 3, we rely on the concept of zero-chain functions, i.e. such ones that a single gradient evaluation makes accessible at most one non-zero coordinate of $x$. By carefully decomposing an appropriate zero-chain function into four components, corresponding to $(f - f_1)$, $(g - g_1)$, $f_1$, $g_1$, and rescaling them to satisfy Assumption 2, we arrive at the following result.

**Theorem 2.** *There exists such $h$, satisfying Assumptions 1, 2, that any algorithm $\mathcal{A}$ (see Definition 1) requires*

$$\Omega\left(\frac{\delta_f}{\varepsilon^2}\right),\ \Omega\left(\frac{\delta_g}{\varepsilon^2}\right)\ \text{calls of } \nabla f,\ \nabla g$$

*to reach an arbitrary $\varepsilon$-solution.*

This result matches the one obtained in Corollary 1. Thus, Algorithm 1 appropriately separates the oracle complexities and enjoys optimal theoretical guarantees.

### 6.2 COORDINATE-BASED STRUCTURED HETEROGENEITY

---

**Algorithm 2** C-HASCA

**Input:** initial points $x^0, w^0 \in \mathbb{R}^d$, number of iterations $T$
**Hyperparameters:** step size $\theta > 0$, probability of full aggregation $p \in (0, 1)$
  1: **for** $t = 0, 1, \ldots, T-1$ **do**
  2:     $e^t = \left[\nabla_x(h - h_1)^\top(w^t), \nabla_y(h - h_1)^\top(x^t, y^t)\right]^\top$
  3:     $(x^{t+1}, y^{t+1}) = \arg\min_{z \in \mathbb{R}^{d_x} \times \mathbb{R}^{d_y}} [B_\theta^t(z)]$, where

$$B_\theta^t(z) = \langle e^t, z \rangle + \frac{1}{2\theta}\|z - (x^t, y^t)\|^2 + h_1(z)$$

  4:     $\omega^{t+1} = \begin{cases} (x^{t+1}, y^{t+1}) & \text{with probability } p \\ \omega^t & \text{with probability } 1 - p \end{cases}$
  5: **end for**
  6: **Output:** $(x^T, y^T)$

---

In this section, we consider the non-convex problem 4 under Assumption 3. C-HASCA (Algorithm 2) is based on the same idea as Algorithm 1. The key difference lies in the approach to approximation of $\nabla(h - h_1)$ (Line 2). Since a block-coordinate formulation with $\delta_x < \delta_y$ is explored, we utilize

the reference point $w^t$ (Line 4) to reduce the frequency of exchanging statistics within the block of parameters $x$.

Unlike the previous setting, coordinate-based structured heterogeneity poses no potential for improvement in the sense of rounds or client-server vector exchanges. Indeed, calling one of the oracles less frequently does not reduce the number of clients involved in synchronization. However, there is a change in amount of bits transmitted from devices to the server. In this regard, we exploit the information-based metric to analyze efficiency of Algorithm 2.

**Corollary 2.** *Suppose Assumptions 1, 3. Consider* $\theta \leq \min\left\{1/8(\delta_x+\delta_y+2\delta_{xy}), p/16\sqrt{2}\max\{\delta_x,\delta_{xy}\}\right\}$. *Algorithm 2 with* $p = (\delta_x+\delta_{xy})/(\delta_x+\delta_y+\delta_{xy})$ *requires*

$$\mathcal{O}\left(\frac{d_x\delta_x}{\varepsilon^2} + \frac{d_y\delta_y}{\varepsilon^2} + \frac{(d_x+d_y)\delta_{xy}}{\varepsilon^2}\right) \text{ coordinates}$$

*to reach an arbitrary $\varepsilon$-solution.*

As mentioned earlier, the main competitor of our methods is `ProxyProx` (Woodworth et al., 2023). Under coordinate-based structured heterogeneity, this scheme requires communicating $\mathcal{O}\left((d_x+d_y)(\delta_x+\delta_y+\delta_{xy})/\varepsilon^2\right)$ bits to converge. Taking $\delta_x < \delta_y$ and $d_y < d_x$ into account, one can note that such a rate is $\mathcal{O}\left(1 + d_x\delta_y/(d_y\delta_x+d_y\delta_{xy})\right)$ times worse than the result of `C-HASCA`. The greater the imbalance between blocks of parameters and the larger the homogeneous component, the more pronounced the advantage of our method becomes.

### 6.2.1 LOWER BOUNDS

Same as for previous case, we present lower bounds for the non-convex problem 4. We use similar set of techniques to construct the worst function. Below, we present the corresponding result.

**Theorem 3.** *There exists such h, satisfying Assumptions 1, 2, that any algorithm $\mathcal{A}$ (see Definition 1) requires to transmit*

$$\Omega\left(\frac{d_x\delta_x}{\varepsilon^2} + \frac{d_y\delta_y}{\varepsilon^2} + \frac{(d_x+d_y)\delta_{xy}}{\varepsilon^2}\right) \text{ coordinates}$$

*to reach an arbitrary $\varepsilon$-solution when $\delta_{xy} < \delta_x$.*

## 7 NUMERICAL EXPERIMENTS

To support theoretical findings, we evaluate the efficiency of `HASCA` (Algorithm 1) and `C-HASCA` (Algorithm 2) in terms of oracle complexity. To provide a comparison, we run several optimizers: `ProxyProx` (Woodworth et al., 2023), a standard method commonly used as a basis for developing new algorithms handling data similarity; `Accelerated ExtraGradient` (Kovalev et al., 2022), a scheme enjoying optimal dependence on $\delta_h$ (see equation 2) for convex objectives; `Adam` (Kingma and Ba, 2014), an algorithm that performs as a strongest baseline while training complex neural networks; `FedProx` (Li et al., 2020) and `SCAFFOLD` (Karimireddy et al., 2020), traditional federated learning methods that are conceptually close to the proposed approach.

One of the possible concerns regarding Algorithms 1, 2 is inability to exactly solve the subproblem (Line 3). However, in our experiments, $\|\nabla h_1^0\|/\|\nabla h_1^k\| \approx 0.1$ was usually enough to achieve convergence.

### 7.1 EXPERIMENTS WITH ALGORITHM 1

In this subsection, we use *ResNet-18* (Meng et al., 2019) to classify *CIFAR-10* (Krizhevsky et al., 2009). This is a 10-class dataset containing $50,000$ training and $10,000$ test samples.

**Experimental setup.** The server holds $15,000$ training samples, while the remaining $35,000$ are distributed across 70 clients. We solve the problem 3 with the cross-entropy loss function. The training data is split into two groups: some amount of classes belongs to one, and the remaining ones to another. To simulate a scenario with both rare and common data modes, we manually introduce a class distribution shift via a constant $\kappa$. It is defined as the ratio of group-one samples stored on the server to the total size of its local dataset. We include a comprehensive study on robustness to $\kappa$ values. We also note that the first convolution in *ResNet-18* is modified, since the input images have sizes of $32 \times 32$ (see the attached code).

**Tuning of Algorithm 1.** To effectively navigate the complex loss landscape of a neural network, we design a practical version of Algorithm 1. To maintain computational efficiency, each local device processes only a batch of its samples per iteration. To align with the theory and approximate full gradients on average, we smooth $e^t$ with its history as a running average. Moreover, obsolescence of the reference $w^t$ (see Line 2 of Algorithm 1) becomes increasingly significant as the optimization approaches the optimum, limiting accuracy to around 60%. Thus, another crucial modification involves introducing a parameter $\alpha^t$ to control the influence of server's descent directions. To achieve competitive performance, we decrease $\alpha^t$ from 1 to 0.2 during training. To sum up, $A_\theta^t$ from Line 3 is modified as follows:

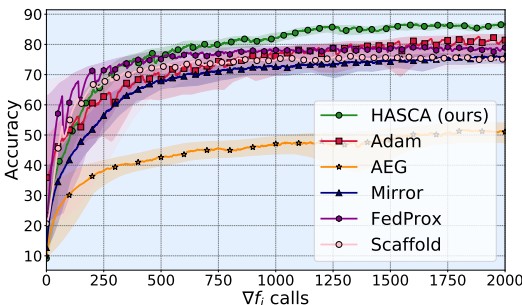

Figure 1: Comparison of `HASCA` to competitors. 5 classes are represented on server with $\kappa = 0.8$. Initial step size $\theta^0$ is set to 0.3. Number of calls of clients from $M_f$ is taken as a criterion.

$$A_\theta^t(x) = \langle \alpha^t m^t, x \rangle + \frac{1}{2\theta^t}\|x - x^t\|^2 + h_1(x),$$

where $m^t = \beta m^{t-1} + (1-\beta)e^t$, $\theta^t$ decreases 30% of the initial value, $\beta$ is set to 0.9.

**Discussion of the results.** Figure 1 illustrates that incomplete coverage of training data by the server does not harm the quality of approximation. When half of the classes are poorly represented on the server, classic distributed methods experience degradation caused by the

| Missing classes | 50% | 60% | 70% | 80% |
|---|---|---|---|---|
| Test accuracy | 86.5% | 85.8% | 83.7% | 84.3% |

Table 1: Test accuracies depending on the proportion of classes poor represented on server. `HASCA` is used as an optimizer. $\kappa$ is set to 0.8.

presence of a poorly conditioned loss component (the numbers of $\nabla f$ and $\nabla g$ coincide). Our method is free of this drawback and can maintain performance even if the server does not have much knowledge. Moreover, the ablation study demonstrates robustness to further reduction in the number of classes represented on the server. Table 1 shows that acceptable quality is maintained even when 80% of the data modes are represented only on clients. Additional experiments can be found in Appendix.

## 7.2 EXPERIMENTS WITH ALGORITHM 2

Here, we solve the binary classification problem on *Avito*[1] text+image multi-modal dataset.

**Experimental setup.** The server holds 60000 samples, while the remaining 140000 are independently shared between 70 clients. We feed the outputs of *ResNet-18* (Meng et al., 2019) and *BERT* (Devlin et al., 2019) into a trainable classification layer. We numerically observe $\delta_x \approx 500$ and $\delta_y \approx 250000$.

**Tuning of Algorithm 2.** To make a single run faster, we choose $p = 0.04$. Algorithm 2 is modified with `Adam`-like momentum and adaptive stepsize with default parameters $\beta_1 = 0.9$, $\beta_2 = 0.999$. We do not use any scheduler in this experiment.

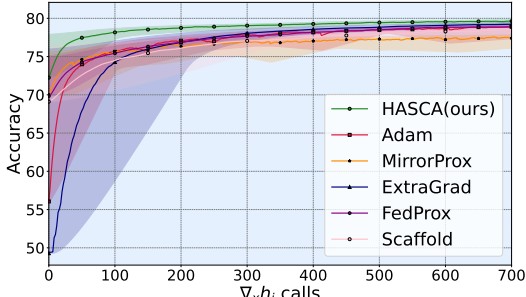

Figure 2: Comparison of `C-HASCA` to mentioned competitors. Number of $\nabla_x h_i$ is taken as a criterion. $\theta^0$ is set to 0.001 without scheduling.

**Discussion of the results.** Figure 2 demonstrates the number of evaluations of a well-conditioned oracle $\nabla_x h$. The communication efficiency of competing methods suffers due to image processing. At the same time, it is possible to evaluate the gradient based on parameters corresponding to textual modality much less frequently by using `C-HASCA`.

[1] https://www.kaggle.com/datasets/antonoof/avito-data

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

APPENDIX

CONTENTS

To ensure reproducibility, we attach the code: `https://anonymous.4open.science/r/hasca-031F/README.md`

## A ADDITIONAL EXPERIMENTS

### A.1 ALGORITHM 1, $\kappa = 0.8$

Recall that $\delta_f < \delta_g$. In this experiment, the server holds only 20% (12000) of the samples associated with $g$, which makes the heterogeneity bound $\delta_g$ approximately 4 times more than for $f$. For existing similarity-aware techniques, this leads to an increased number of oracle calls for both components (see Figure 3). In contrast, our approach accounts for this shift in mode heterogeneity and allows one of the gradients to be evaluated less frequently than the other while maintaining training quality. Notably, our method not only demonstrates faster loss decrease on the training set but also achieves superior accuracy. This highlights its potential for practical extensions that adapt well to the highly non-convex landscape of neural networks. In terms of the number of evaluations of the $g$ component, which the server approximates poorly, our method remains competitive, which meets our theoretical guarantees, mentioned in the Corollary 1.

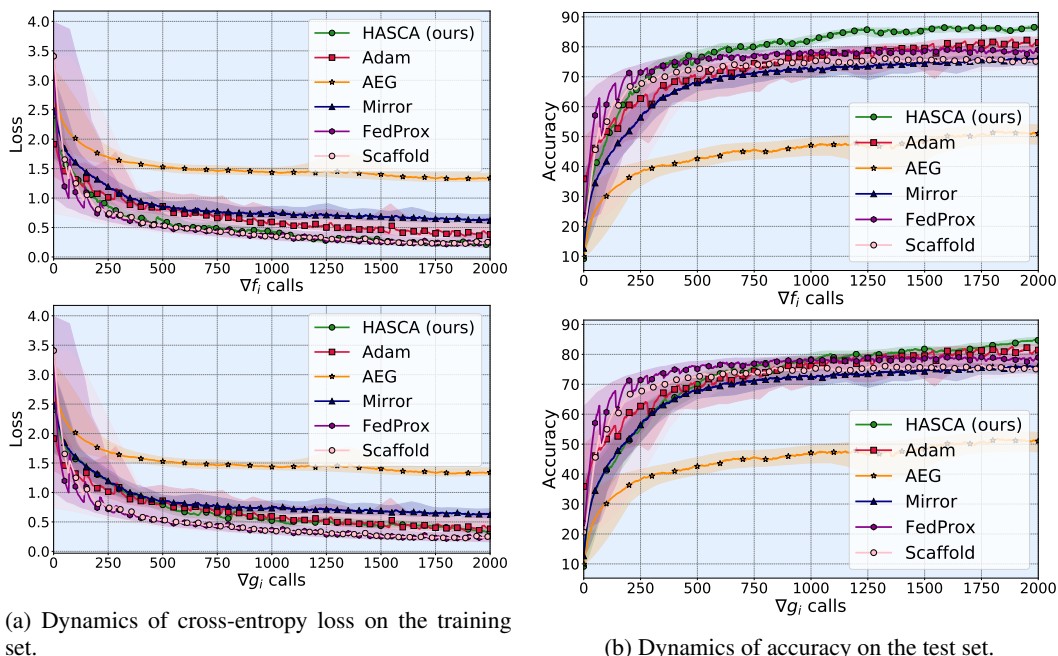

(a) Dynamics of cross-entropy loss on the training set.

(b) Dynamics of accuracy on the test set.

Figure 3: Comparison of HASCA to mentioned competitors. Number of synchronizations with clients from $M_f$ is taken as a criterion. $\kappa$ is set to 0.8, and initial step size $\theta^0$ is set to 0.3

### A.2 ALGORITHM 1, ADDITIONAL $\kappa$'S

In the main part of our work, we focus on moderate setting, which best captures the essence of the proposed approach. Indeed, for $\kappa = 0.8$, Adam still maintains strong performance, while the gap between classic similarity-accounted schemes and our approach becomes noticeable. Nevertheless, for the sake of methodological completeness, we also conduct experiments in two extreme cases.

**Experiments with $\kappa = 0.6$.** This setup assumes that the class imbalance on the server is minimal. We test this scenario to ensure that the method does not become ineffective as $\kappa$ decreases. Figure 4a shows that with low values of $\kappa \approx 0.5$ HASCA (Algorithm 1) and Mirror Prox share the same quality.

**Experiments with $\kappa = 0.95$.** This extremely heterogeneous scenario is designed to demonstrate the method's robustness to increasing distribution shift and to explore its full potential advantage

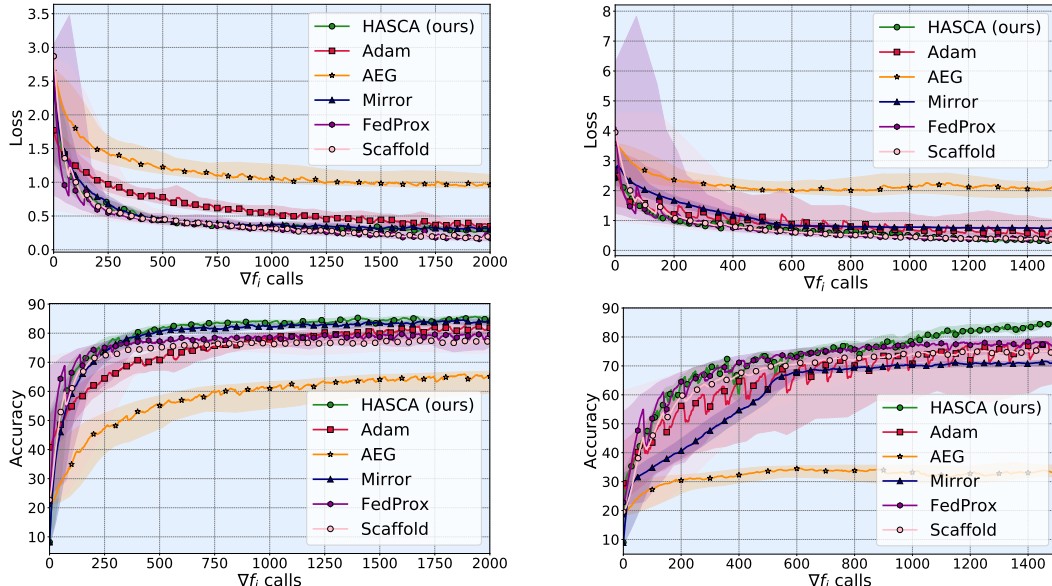

(a) Dynamics of train cross-entropy loss and test accuracy, $\kappa = 0.6$.

(b) Dynamics of train cross-entropy loss and test accuracy, $\kappa = 0.95$.

Figure 4: Comparison of HASCA to mentioned competitors. Number of synchronizations with clients from $M_f$ is taken as a criterion. Initial $\theta^0$ is set above 0.5 and quickly decreased to 0.05 and 0.001, respectively

over current state-of-the-art approaches. Figure 4b shows that with extreme $\kappa = 0.95$ only HASCA (Algorithm 1) has the ability to achieve *ResNet-18*'s accuracy limits.

## A.3 ALGORITHM 1, ROBUSTNESS TO $p$

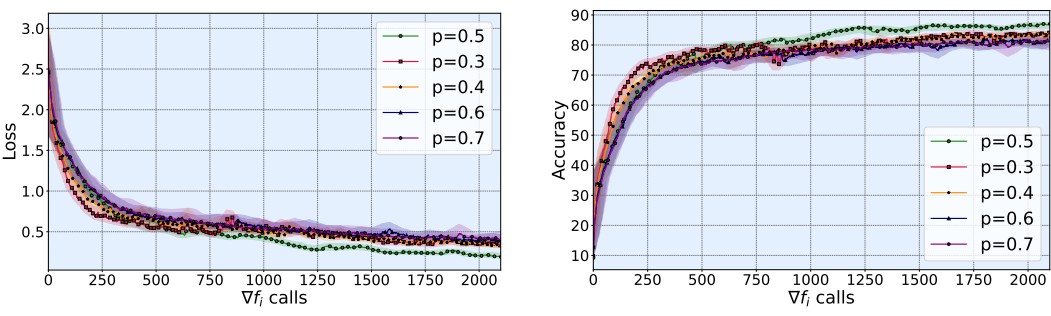

Figure 5: Robustness of HASCA to changes in $p$. $\kappa$ is set to $0.8$, and initial step size $\theta^0$ is set to $0.3$

In this section, we study the robustness of our schemes to variations in the probability of full aggregation $p$. Theorem 1 provides the optimal value $p = \delta_f/(\delta_f + \delta_g)$. Although this quantity cannot be computed exactly in practice, it can be evaluated via approximations $\delta_f \sim 1/\sqrt{N_f}$ and $\delta_g \sim 1/\sqrt{N_g}$ (Hendrikx et al., 2020). Here, $N_f$ and $N_g$ denote the number of samples in the server's dataset associated with $f$ and $g$, respectively. Hence, the initialization $p^0$ for tuning should be chosen according to $p^0 = \sqrt{N_g}/(\sqrt{N_f} + \sqrt{N_g})$.

If $\kappa$ is set to $0.8$, the optimal choice is $p = 0.5$. Figure 5 shows that even under substantial deviations from the tuned value of $p$, the performance of Algorithm 1 does not deteriorate drastically. The degradations shown in Figure 5 can be explained by the fact that when $p$ is too small, the server communicates with $M_g$ less frequently than would be appropriate given the similarity of optimiza-

tion landscapes. As a result, although Algorithm 1 with $p = 0.3$ converges faster in the initial iterations, it slows down once the server's knowledge becomes insufficient for the model to continue successfully learning the underlying dependencies. Conversely, when $p$ is too large, we observe a slowdown caused by overly frequent full aggregations.

### A.4 ALGORITHM 1, ROBUSTNESS TO CLASS IMBALANCE

In the main part of this work, we considered that server's data represents the half of all classes well. Such a setup is fairly mild. In this section, we examine how the quality of learning changes when less than $50\%$ of classes can be well approximated by the server. We conduct the ablation study with $\kappa = 0.8$.

As mentioned earlier, our algorithm shows strong robustness to the decreasing number of well-known classes stored by the server. Actually, such cases are even more heterogeneous than $\kappa = 0.95$ case. Moreover, even with $q = 20\%$, when each server batch contains only $2.5\%$ of each client class, no quality drops are observed.

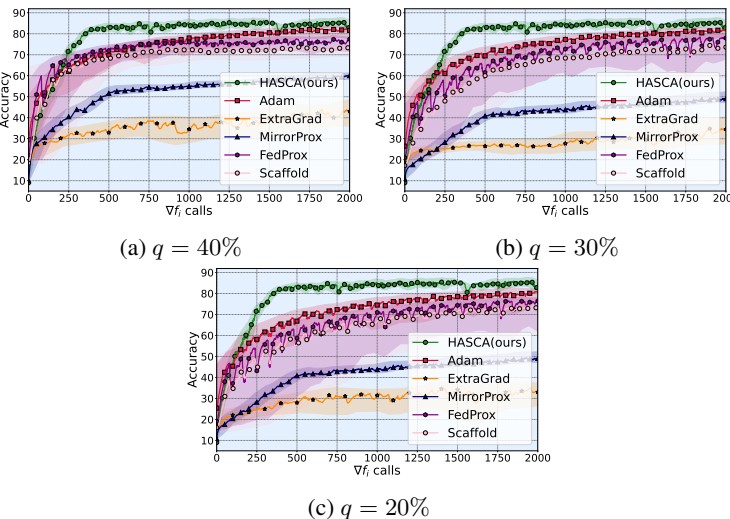

(a) $q = 40\%$        (b) $q = 30\%$

(c) $q = 20\%$

Figure 6: Performance of HASCA with different proportion $q$ of well-approximated classes. Number of synchronizations with clients from $M_f$ is taken as a criterion. $\kappa$ is set to $0.8$

### A.5 ALGORITHM 1, RUNTIME

Since Algorithms 1, 2 involve inner minimization, there is the question of their runtime compared to competitors. In this section, we conduct a numerical study of execution time in federated networks consisting of 10 devices with $\kappa = \{0.6, 0.8, 0.95\}$. The network bandwidth for client-server communication is approximately 25Mbps. The inner minimization is solved by the server in 8–14 seconds.

| Time (Sec) | 1250 | 2500 | 5000 | 7500 | 10000 | 12500 | 15000 |
|---|---|---|---|---|---|---|---|
| ExtraGrad | 32.77% | 40.75% | 48.92% | 55.53% | 57.68% | 58.67% | 61.30% |
| MirrorProx | 53.26% | 64.4% | 75.33% | 78.58% | 81.59% | 81.91% | 82.17% |
| Adam | 53.92% | 62.18% | 69.36% | 73.44% | 73.94% | 74.77% | 79.69% |
| FedProx | **70.91%** | **72.56%** | 76.22% | 77.02% | 77.35% | 77.85% | 78.62% |
| SCAFFOLD | 68.91% | 70.26% | 74.42% | 75.06% | 75.84% | 77.22% | 76.45% |
| HASCA | 59.94% | 68.53% | **76.76%** | **80.16%** | **82.21%** | **82.96%** | **83.33%** |

Table 2: Comparison of HASCA with competitors in terms of runtime. $\kappa$ is set to $0.6$. At each timestamp, we highlight the **best** test accuracy.

| Time (Sec) | 1250 | 2500 | 5000 | 7500 | 10000 | 12500 | 15000 |
|---|---|---|---|---|---|---|---|
| ExtraGrad | 30.45% | 34.42% | 40.13% | 43.10% | 45.47% | 45.70% | 46.20% |
| MirrorProx | 37.27% | 47.84% | 57.79% | 65.86% | 69.26% | 71.17% | 72.04% |
| Adam | 50.54% | 60.83% | 66.71% | 70.61% | 71.61% | 73.42% | 79.54% |
| FedProx | **64.66%** | **67.16%** | **73.62%** | **75.54%** | 76.71% | 76.98% | 77.71% |
| SCAFFOLD | 61.86% | 65.62% | 71.82% | 71.85% | 72.98% | 73.91% | 74.24% |
| HASCA | 42.93% | 55.39% | 69.43% | 73.60% | **77.22%** | **79.02%** | **81.02%** |

Table 3: Comparison of HASCA with competitors in terms of runtime. $\kappa$ is set to $0.8$. At each timestamp, we highlight the **best** test accuracy.

| Time (Sec) | 1250 | 2500 | 5000 | 7500 | 10000 | 12500 | 15000 |
|---|---|---|---|---|---|---|---|
| ExtraGrad | 26.95% | 28.54% | 30.79% | 33.08 | 33.82% | 34.94% | 33.44% |
| MirrorProx | 33.2% | 37.63% | 45.89% | 55.25% | 65.94% | 68.43% | 69.71% |
| Adam | 40.75% | 52.85% | 56.59% | 62.53% | 68.99% | 69.08% | 75.68% |
| FedProx | **57.26%** | **61.67%** | **68.05%** | **71.92%** | **73.65%** | 74.82% | 75.53% |
| SCAFFOLD | 54.06% | 58.51% | 65.25% | 68.50% | 71.38% | 72.41% | 72.93% |
| HASCA | 42.80% | 58.44% | 66.10% | 63.19% | 71.13% | **75.90%** | **75.69%** |

Table 4: Comparison of HASCA with competitors in terms of runtime. $\kappa$ is set to $0.95$. At each timestamp, we highlight the **best** test accuracy.

Tables 2-4 reveal several observations. First, as $\kappa \to 0.5$, similarity-based methods (MirrorProx, ExtraGrad) tend to outperform Adam, SCAFFOLD and FedProx. Moreover, the performance of MirrorProx becomes close to Algorithm 1, since $\delta_f + \delta_g$ decreases while $\delta_f$ increases. This is consistent with the fact that Algorithm 1 coincides with MirrorProx when $\kappa = 0.5$. Second, as $\kappa \to 1$, the gap between Algorithm 1 and other Mirror-like schemes increases, since the sum $\delta_f + \delta_g$ grows while $\delta_f$ decreases. This behavior is also consistent with the analysis and explains the superior performance of our method in heterogeneous federated learning scenarios.

### A.6 Scalability of Algorithm 1

In this section, we discuss the scalability of our approach. We use Food101 Bossard et al. (2014) with FasterViT (Hatamizadeh et al., 2023) for fine-tuning, providing a complex benchmark for comparing Algorithm 1.

| # of $\nabla f$ calls | 500 | 1000 | 1500 | 2000 | 2250 |
|---|---|---|---|---|---|
| Adam | 46.53% | 62.46% | 68.72% | 73.90% | 74.43% |
| HASCA | **47.40%** | **63.74%** | **71.26%** | **74.85%** | **75.40%** |

Table 5: Comparison of HASCA with Adam. $\kappa$ is set to $0.8$. At each stamp, we highlight the **best** test accuracy.

Table 5 demonstrates that our method retains its properties when transitioning from training a fairly simple model with $11.5$M parameters to fine-tuning the complex *ViT*-270M model.

## B Proof of Lemma 1

**Lemma 2 (Lemma 1).** *Suppose Assumptions 1, 2 hold. Then, for Algorithm 1 it implies*

$$\mathbb{E}_{e^{t+1}}\left[\|e^{t+1} - \nabla(h - h_1)(x^{t+1})\|^2\right] \leq \left(1 - \frac{p}{2}\right)\|e^t - \nabla(h - h_1)(x^t)\|^2 + \frac{2}{p}\delta_f^2\|x^{t+1} - x^t\|^2.$$

*Proof.* Let us note that the update of $e^t$ (see Line 2 of Algorithm 1) can be rewritten as

$$e^t = \begin{cases} \nabla(h - h_1)(x^t), & \text{with probability } p \\ e^{t-1} + \nabla(g - g_1)(x^t) - \nabla(g - g_1)(x^{t-1}), & \text{with probability } 1 - p \end{cases}.$$

In this proof, we exploit this equivalent representation of $e^t$. We take

$$\nabla(g - g_1) - \nabla(h - h_1) = \nabla(f - f_1)$$

into account and write

$$\mathbb{E}_{e^{t+1}}\left[\|e^{t+1} - \nabla(h - h_1)(x^{t+1})\|^2\right] = (1-p)\|e^t - \nabla(g - g_1)(x^t) - \nabla(f - f_1)(x^{t+1})\|^2.$$

Adding and subtracting $\nabla(h - h_1)(x^t)$, we derive

$$\mathbb{E}_{e^{t+1}}\left[\|e^{t+1} - \nabla(h - h_1)(x^{t+1})\|^2\right] \leq (1-p)(1+c)\|e^t - \nabla(h - h_1)(x^t)\|^2$$
$$+ \left(1 + \frac{1}{c}\right)\|\nabla(f - f_1)(x^t) - \nabla(f - f_1)(x^{t+1})\|^2.$$

Appplying Assumption 2 to the right hand of this inequality, we get

$$\mathbb{E}_{e^{t+1}}\left[\|e^{t+1} - \nabla(h - h_1)(x^{t+1})\|^2\right] \leq (1-p)(1+c)\|e^t - \nabla(h - h_1)(x^t)\|^2$$
$$+ \left(1 + \frac{1}{c}\right)\delta_f^2\|x^{t+1} - x^t\|^2.$$

To obtain a linear decrease in approximation drift, we choose $c = \frac{p}{2}$ and arrive at

$$\mathbb{E}_{e^{t+1}}\left[\|e^{t+1} - \nabla(h - h_1)(x^{t+1})\|^2\right] \leq \left(1 - \frac{p}{2}\right)\|e^t - \nabla(h - h_1)(x^t)\|^2 + \frac{2}{p}\delta_f^2\|x^{t+1} - x^t\|^2.$$

This concludes the proof of Lemma 1. □

## C    PROOF OF THEOREM 1

**Theorem 4** (Theorem 1). *Suppose Assumptions 1, 2 hold. Consider $\theta \leq \min\left\{1/8(\delta_f+\delta_g), p/8\sqrt{2}\delta_f\right\}$. Then, Algorithm 1 requires*

$$\mathcal{O}\left(\frac{\delta_f + \delta_g}{\varepsilon^2} + \frac{\delta_f}{p\varepsilon^2}\right) \text{ iterations}$$

*to achieve an arbitrary $\varepsilon$-solution, where $\varepsilon^2 = \mathbb{E}\left[\left\|\frac{1}{T}\sum_{t=1}^{T}\nabla h\left(x^t\right)\right\|^2\right].$*

*Proof.* Since our goal is to provide a heterogeneity-accounted analysis, we can not rely on the smoothness of the objective, which serves as the basic descent lemma in nonconvex analysis. Instead, we derive its analogue. Let us start with

$$h(x^{t+1}) - h(x^t) = \int_0^1 dh(x^t + \tau(x^{t+1} - x^t))d\tau = \int_0^1 \langle\nabla h(x^t + \tau(x^{t+1} - x^t)), x^{t+1} - x^t\rangle$$

and the same

$$h_1(x^{t+1}) - h_1(x^t) = \int_0^1 \langle\nabla h_1(x^t + \tau(x^{t+1} - x^t)), x^{t+1} - x^t\rangle.$$

Summing up this inequalities, we obtain

$$h(x^{t+1}) - h(x^t) = \int_0^1 \langle\nabla(h - h_1)(x^t + \tau(x^{t+1} - x^t)), x^{t+1} - x^t\rangle d\tau \qquad (5)$$
$$+ h_1(x^{t+1}) - h_1(x^t).$$

Since $x^{t+1}$ is the minimum of $A_\theta^t$ defined in Line 3, we have

$$h_1(x^{t+1}) - h(x^t) \leq -\langle e^t, x^{t+1} - x^t\rangle - \frac{1}{2\theta}\|x^{t+1} - x^t\|^2.$$

Substituting it into equation 5 and applying $\nabla(h - h_1)(x^t)$ as a smart zero, we get

$$h(x^{t+1}) - h_1(x^t) \leq -\frac{1}{2\theta}\|x^{t+1} - x^t\|^2$$
$$+ \int_0^1 \langle\nabla(h - h_1)(x^t + \tau(x^{t+1} - x^t)) - \nabla(h - h_1)(x^t), x^{t+1} - x^t\rangle d\tau$$
$$+ \langle\nabla(h - h_1)(x^t) - e^t, x^{t+1} - x^t\rangle.$$

After applying Young's inequality, this turns into

$$h(x^{t+1}) - h(x^t) \leq -\frac{1}{2\theta}\|x^{t+1} - x^t\|^2$$

$$+ \int_0^1 \langle \nabla(h - h_1)(x^t + \tau(x^{t+1} - x^t)) - \nabla(h - h_1)(x^t), x^{t+1} - x^t \rangle \mathrm{d}\tau \quad (6)$$

$$+ \frac{\alpha}{2}\|\nabla(h - h_1)(x^t) - e^t\|^2 + \frac{1}{2\alpha}\|x^{t+1} - x^t\|^2.$$

Let us consider the integral separately. We have

$$\int_0^1 \langle \nabla(h - h_1)(x^t + \tau(x^{t+1} - x^t)) - \nabla(h - h_1)(x^t), x^{t+1} - x^t \rangle \mathrm{d}\tau$$

$$\leq \int_0^1 \|\nabla(h - h_1)(x^t + \tau(x^{t+1} - x^t)) - \nabla(h - h_1)(x^t)\|\|x^{t+1} - x^t\|\mathrm{d}\tau$$

$$\leq \int_0^1 (\delta_f + \delta_g)\|x^t + \tau(x^{t+1} - x^t) - x^t\|\|x^{t+1} - x^t\|\mathrm{d}\tau$$

$$= \int_0^1 \tau(\delta_f + \delta_g)\|x^{t+1} - x^t\|^2\mathrm{d}\tau$$

$$= (\delta_f + \delta_g)\|x^{t+1} - x^t\|^2.$$

We substitute this into equation 6 with $\alpha = 2\theta$ and derive

$$h(x^{t+1}) - h(x^t) \leq \left(-\frac{1}{4\theta} + \delta_f + \delta_g\right)\|x^{t+1} - x^t\|^2 + \theta\|e^t - \nabla(h - h_1)(x^t)\|^2. \quad (7)$$

Let us consider $\Phi^t = h(x^t) - h(x^*) + A\|e^t - \nabla(h - h_1)(x^t)\|^2$ as a potential function. We begin with writing a recursion

$$\Phi^{t+1} = [h(x^t) - h(x^*)] + [h(x^{t+1}) - h(x^t)] + A\|e^{t+1} - \nabla(h - h_1)(x^{t+1})\|^2$$

$$\leq [h(x^t) - h(x^*)] + \theta\|e^t - \nabla(h - h_1)(x^t)\|^2 + \left(-\frac{1}{4\theta} + \delta_f + \delta_g\right)\|x^{t+1} - x^t\|^2$$

$$+ A\|e^{t+1} - \nabla(h - h_1)(x^{t+1})\|^2,$$

where the last transition exploits equation 7. Next, we apply Lemma 1 and derive

$$\mathbb{E}_{e^{t+1}}\left[\Phi^{t+1}\right] = [h(x^t) - h(x^*)] + [h(x^{t+1}) - h(x^t)] + A\mathbb{E}_{e^{t+1}}\left[\|e^{t+1} - \nabla(h - h_1)(x^{t+1})\|^2\right]$$

$$\leq [h(x^t) - h(x^*)] + \theta\|e^t - \nabla(h - h_1)(x^t)\|^2 + \left(-\frac{1}{4\theta} + \delta_f + \delta_g\right)\|x^{t+1} - x^t\|^2$$

$$+ A\left(1 - \frac{p}{2}\right)\|e^t - \nabla(h - h_1)(x^t)\| + \frac{2A}{p}\delta_f^2\|x^{t+1} - x^t\|^2.$$

After grouping terms, we arrive at

$$\mathbb{E}_{e^{t+1}}\left[\Phi^{t+1}\right] \leq \Phi^t + \left(-\frac{1}{4\theta} + \delta_f + \delta_g + \frac{2A}{p}\delta_f^2\right)\|x^{t+1} - x^t\|^2$$

$$+ \left(\theta - \frac{Ap}{2}\right)\|e^t - \nabla(h - h_1)(x^t)\|. \quad (8)$$

Let us deal with the first term of equation 8. First, let us note that Line 3 of Algorithm 1 implies

$$0 = e^t + \nabla(h_1 - h)(x^{t+1}) + \nabla h(x^{t+1}) + \frac{x^{t+1} - x^t}{\theta}$$

$$= [e^t - \nabla(h - h_1)(x^t)] + [\nabla(h - h_1)(x^{t+1}) - \nabla(h - h_1)(x^t)] + \nabla h(x^{t+1}) + \frac{x^{t+1} - x^t}{\theta}.$$

This implies

$$\|x^{t+1} - x^t\|^2 = \theta^2\|[e^t - \nabla(h - h_1)(x^t)] + [\nabla(h - h_1)(x^{t+1}) - \nabla(h - h_1)(x^t)] + \nabla h(x^{t+1})\|^2.$$

Next, we employ the trick. We write

$$\|\nabla h(x^{t+1})\|^2 = \|\nabla h(x^{t+1}) + [e^t - \nabla(h-h_1)(x^t)] + [\nabla(h-h_1)(x^{t+1}) - \nabla(h-h_1)(x^t)]$$
$$- [e^t - \nabla(h-h_1)(x^t)] - [\nabla(h-h_1)(x^{t+1}) - \nabla(h-h_1)(x^t)]\|^2$$
$$\leq 3\|\nabla h(x^{t+1}) + [e^t - \nabla(h-h_1)(x^t)] + [\nabla(h-h_1)(x^{t+1}) - \nabla(h-h_1)(x^t)]\|^2$$
$$+ 3\|e^t - \nabla(h-h_1)(x^t)\|^2 + 3\|\nabla(h-h_1)(x^{t+1}) - \nabla(h-h_1)(x^t)\|^2.$$

This implies

$$\|\nabla h(x^{t+1}) + [e^t - \nabla(h-h_1)(x^t)] + [\nabla(h-h_1)(x^{t+1}) - \nabla(h-h_1)(x^t)]\|^2$$
$$\geq \frac{1}{3}\|\nabla h(x^{t+1})\|^2 - \|e^t - \nabla(h-h_1)(x^t)\|^2 - \|\nabla(h-h_1)(x^{t+1}) - \nabla(h-h_1)(x^t)\|^2$$
$$\geq \frac{1}{3}\|\nabla h(x^{t+1})\|^2 - \|e^t - \nabla(h-h_1)(x^t)\|^2 - (\delta_f + \delta_g)^2\|x^{t+1} - x^t\|^2.$$

Thus, there is a lower bound on the update:

$$\|x^{t+1} - x^t\|^2 \geq \frac{\theta^2}{3}\|\nabla h(x^{t+1})\|^2 - \theta^2\|e^t - \nabla(h-h_1)(x^t)\|^2 - \theta^2(\delta_f + \delta_g)^2\|x^{t+1} - x^t\|^2.$$

After rearranging terms, we get

$$\left(1 - \theta^2(\delta_f + \delta_g)^2\right)\|x^{t+1} - x^t\|^2 \geq \frac{\theta^2}{3}\|\nabla h(x^{t+1})\|^2 - \theta^2\|e^t - \nabla(h-h_1)(x^t)\|^2.$$

Due to the choice of parameters outlined in Theorem 1, we have $\theta \leq 1/2(\delta_f+\delta_g)$ which implies

$$1 - \theta^2(\delta_f + \delta_g)^2 \geq 1 - \frac{1}{4} = \frac{3}{4}.$$

Thus, we have

$$\|x^{t+1} - x^t\|^2 \geq \frac{4\theta^2}{9}\|\nabla h(x^{t+1})\|^2 - \frac{4\theta^2}{3}\|e^t - \nabla(h-h_1)(x^t)\|^2. \qquad (9)$$

Comparing equation 8 and equation 9, we observe that the variance could be included in the resulting inequality with negative sign only if $A < 2\theta/p$. We choose $A = 4\theta/p$. This means that equation 8 transforms into

$$\mathbb{E}_{e^{t+1}}\left[\Phi^{t+1}\right] \leq \Phi^t + \left(-\frac{1}{4\theta} + \delta_f + \delta_g + \frac{8\theta\delta_f^2}{p^2}\right)\|x^{t+1} - x^t\|^2 - \theta\|e^t - \nabla(h-h_1)(x^t)\|.$$

To substitute equation 9, the first term of this inequality should be negative. Let us show that it is actually negative because of choice of $\theta$ (see Theorem 1). On the one hand, $\theta \leq 1/8(\delta_f+\delta_g)$, which implies $(\delta_f + \delta_g) \leq 1/8\theta$. On the other hand, $\theta \leq p/8\sqrt{2}\delta_f$, which implies $8\theta\delta_f^2/p^2 \leq 1/16\theta$. Thus, we have

$$-\frac{1}{4\theta} + \delta_f + \delta_g + \frac{8\theta\delta_f^2}{p^2} \leq -\frac{1}{16\theta} < 0.$$

Thus, we can apply equation 9 and deduce

$$\mathbb{E}_{e^{t+1}}\left[\Phi^{t+1}\right] \leq \Phi^t - \frac{\theta}{36}\|\nabla h(x^{t+1})\|^2 + \left(\frac{1}{12} - 1\right)\theta\|e^t - \nabla(h-h_1)(x^t)\|^2,$$

which implies

$$\mathbb{E}_{e^{t+1}}\left[\|\nabla h(x^{t+1})\|^2\right] \leq \frac{36}{\theta}[\Phi^t - \Phi^{t+1}].$$

Taking full expectation and accounting for $e^0 = \nabla(h-h_1)(x^0)$, we obtain

$$\mathbb{E}\left[\left\|\frac{1}{T}\sum_{t=1}^T \nabla h(x^t)\right\|^2\right] \leq \frac{36[h(x^0) - h(x^*)]}{\theta} = \mathcal{O}\left(\left(\frac{\delta_f + \delta_g}{T} + \frac{\delta_f}{pT}\right)[h(x^0) - h(x^*)]\right).$$

This concludes the proof of Theorem 1. $\qquad \square$

## D    PROOF OF COROLLARY 1

**Corollary 3** (Corollary 1). *Consider the conditions of Theorem 1. Algorithm 1 with $p = \delta_f/(\delta_f+\delta_g)$ requires*

$$\mathcal{O}\left(\frac{\delta_f}{\varepsilon^2}\right), \ \mathcal{O}\left(\frac{\delta_g}{\varepsilon^2}\right) \ calls$$

*of $\nabla f$, $\nabla g$, respectively, to reach an arbitrary $\varepsilon$-solution*

*Proof.* In Theorem 1 we have established that Algorithm 1 requires $\mathcal{O}\left((\delta_f+\delta_g)/\varepsilon^2 + \delta_f/p\varepsilon^2\right)$ iterations to converge. Since $\nabla g$ is called at every iteration, its oracle complexity is the same. However, $\nabla f$ is evaluated with probability $p$, i.e. $1/p$ times more rarely on average. Thus, its oracle complexity is $\mathcal{O}\left(p(\delta_f+\delta_g)/\varepsilon^2 + \delta_f/\varepsilon^2\right)$. The choice $p = \delta_f/(\delta_f+\delta_g)$ leads to the desired statement. $\qquad\square$

## E    PROOF OF THEOREM 2

To derive lower bounds for the nonconvex problem 3, we rely on the concept of zero-chain function.

**Definition 2.** *Let us define*

$$prog(x) = \begin{cases} 0, & if \ x = 0 \\ \max_{1 \le j \le d}\{j : [x]_j \ne 0\}, & else \end{cases}.$$

*The function $l$ is called zero-chain, if*

$$prog(\nabla l(x)) \le prog(x) + 1.$$

This means that if the process starts at the point $x = 0$, then after a gradient estimation one can earn at most one non-zero coordinate of $x$. In this section, we work zero-chain functions of the form

$$l(x) = -\Psi(1)\Phi([x]_1) + \sum_{j=2}^{d} \left(\Psi(-[x]_{j-1})\Phi(-[x]_j) - \Psi([x]_{j-1})\Phi([x]_j)\right),$$

where

$$\Psi(z) = \begin{cases} 0, & z \le \frac{1}{2} \\ \exp\left\{1 - \frac{1}{(2z-1)^2}\right\}, & z > \frac{1}{2} \end{cases},$$

$$\Phi(z) = \sqrt{e}\int_{-\infty}^{z} \exp\left\{-\frac{t^2}{2}\right\} dt.$$

It has already been shown by Arjevani et al. (2023) that $l$ satisfies the following properties.

1. $l(x) - \inf_{x \in \mathbb{R}^d} l(x) \le \Delta_0 d$ with $\Delta_0 = 12$ for every $x \in \mathbb{R}^d$;

2. $l(x)$ is $L_0$-smooth with $L_0 = 152$;

3. $\|\nabla l(x)\|_\infty \le G_0$ with $G_0 = 23$;

4. $\forall x \in \mathbb{R}^d : [x]_d = 0 \to \|\nabla l(x)\| \ge 1$.

Moreover, let us define $l_j$ as follows:

$$l_j(x) = l_j([x]_{j-1}, [x]_j) = \begin{cases} -\Psi(1)\Phi([x]_1), & j = 1 \\ \Psi(-[x]_{j-1})\Phi(-[x]_j) - \Psi([x]_{j-1})\Phi([x]_j), & j > 1 \end{cases}.$$

It was shown in (Metelev et al., 2024) that $l_j$ is also $L_0$-smooth for every $j$. Now that the set of functions has been introduced and their properties described, we proceed to the proof of Theorem 1.

**Theorem 5** (Theorem 2). *There exists such $h$, satisfying Assumptions 1, 2, that any algorithm $\mathcal{A}$ (see Definition 1) requires*

$$\Omega\left(\frac{\delta_f}{\varepsilon^2}\right), \ \Omega\left(\frac{\delta_g}{\varepsilon^2}\right) \ calls$$

*of $\nabla f$, $\nabla g$, respectively.*

*Proof.* Let us represent $x \in \mathbb{R}^d$ as $x \in \mathbb{R}^{d_1} \times \mathbb{R}^{d_2}$, where $d_1$ is the odd number. Based on this, we define two sets:

$$S_1 = \{i \in [1, \ldots, d_1 + d_2] : i \mod 2 = 1\},$$
$$S_2 = [1, \ldots, d_1 + d_2] \setminus S_1.$$

Now, we further divide each of these two sets into two more subsets:

$$S_{11} = \{i \in S_1 : i \leq d_1\}, \quad S_{12} = \{i \in S_1 : i \in (d_1, d_1 + d_2]\}$$
$$S_{21} = \{i \in S_2 : i \leq d_1\}, \quad S_{22} = \{i \in S_2 : i \in (d_1, d_1 + d_2]\}.$$

Based on these sets of indices, we define $\tilde{f}$ and $\tilde{g}$:

$$\tilde{f}(x) = -\Psi(1)\Phi([x]_1) + \sum_{j=2}^{d_1} \left( \Psi(-[x]_{j-1})\Phi(-[x]_j) - \Psi([x]_{j-1})\Phi([x]_j) \right),$$

$$\tilde{g}(x) = -\Psi(1)\Phi([x]_{d_1+1}) + \sum_{j=d_1+2}^{d_1+d_2} \left( \Psi(-[x]_{j-1})\Phi(-[x]_j) - \Psi([x]_{j-1})\Phi([x]_j) \right).$$

We share these functions between clients and the server according to index sets in the following way:

$$\tilde{f}_1(x) = \Psi(1)\Phi([x]_1) + \sum_{j \in S_{11}, j \geq 2} \left( \Psi(-[x]_{j-1})\Phi(-[x]_j) - \Psi([x]_{j-1})\Phi([x]_j) \right),$$

$$(\tilde{f} - \tilde{f}_1)(x) = \sum_{j \in S_{21}, j \geq 2} \left( \Psi(-[x]_{j-1})\Phi(-[x]_j) - \Psi([x]_{j-1})\Phi([x]_j) \right),$$

$$\tilde{g}_1(x) = -\Psi(1)\Phi([x]_{d_1+1}) + \sum_{j \in S_{12}, j \geq d_1+2} \left( \Psi(-[x]_{j-1})\Phi(-[x]_j) - \Psi([x]_{j-1})\Phi([x]_j) \right),$$

$$(\tilde{g} - \tilde{g}_1)(x) = \sum_{j \in S_{22}, j \geq d_1+2} \left( \Psi(-[x]_{j-1})\Phi(-[x]_j) - \Psi([x]_{j-1})\Phi([x]_j) \right).$$

These functions are $L_0$-smooth, and we must re-scale them to move into the class of interest. Let us define

$$f_1(x) = \frac{\delta_f^2 C_f^2}{L_0} \tilde{f}_1\left(\frac{x}{C_f}\right), \quad (f - f_1)(x) = \frac{\delta_f^2 C_f^2}{L_0} \tilde{f}_1\left(\frac{x}{C_f}\right),$$

$$g_1(x) = \frac{\delta_g^2 C_g^2}{L_0} \tilde{g}_1\left(\frac{x}{C_g}\right), \quad (g - g_1)(x) = \frac{\delta_g^2 C_g^2}{L_0} \tilde{g}_1\left(\frac{x}{C_g}\right).$$

where $C_f$ and $C_g$ are defined below. One can easily ensure that

$$\|\nabla^2(f - f_1)\| \leq \delta_f, \quad \|\nabla^2(g - g_1)\| \leq \delta_g.$$

Note that since $\tilde{f}$, $\tilde{g}$ do not share any coordinates, the properties mentioned at the beginning of this section hold for each of them separately. Therefore, we have

$$h(0) - \inf_{x \in \mathbb{R}^d} h(x) = [f(0) - \inf_{x \in \mathbb{R}^d} f(x)] + [g(0) - \inf_{x \in \mathbb{R}^d} g(x)] \leq \frac{\delta_f^2 C_f^2}{L_0} \Delta_{\tilde{f}} d_1 + \frac{\delta_g^2 C_g^2}{L_0} \Delta_{\tilde{g}} d_2.$$

Consider the oracle that computes $\nabla f$ with probability $p$ and $\nabla h$ with probability $1$ (see Definition 1). Consider the number of iterations to be fixed and equal to $T$. Then, on average, the algorithm can not make accessible more than $\lfloor pT \rfloor + 1$ and $T + 1$ coordinates corresponding to $f$ and $g$, respectively. Considering $pT \geq 2$, we define

$$d_1 = 1 + \lfloor pT \rfloor < 2pT, \quad d_2 = 1 + T \leq 2T.$$

Next, let us specify the values $C_f, C_g$. We choose them as

$$C_f^2 = \frac{L_0 \Delta_h}{\delta_f \Delta_{\tilde{f}} pT}, \quad C_g^2 = \frac{L_0 \Delta_h}{\delta_g \Delta_{\tilde{g}} T}.$$

Now we use the properties mentioned in the beginning of this section and derive

$$\mathbb{E}\left[\|\nabla h(\hat{x})\|^2\right] \geq \min_{[x]_{d_1}=0, [x]_{d_1+d_2}=0} \|\nabla h(x)\|^2 = \min_{[x]_{d_1}=0} \|\nabla f(x)\|^2 + \min_{[x]_{d_1+d_2}=0} \|\nabla g(x)\|^2$$

$$\geq \frac{\delta_f^2 C_f^2}{L_0^2} \min_{[x]_{d_1}=0} \|\nabla \tilde{f}(x)\|^2 + \frac{\delta_g^2 C_g^2}{L_0^2} \min_{[x]_{d_1+d_2}=0} \|\nabla \tilde{g}(x)\|^2 \geq \frac{\Delta_h}{T}\left(\frac{\delta_f}{p} + \delta_g\right).$$

In terms of convergence to $\varepsilon$-solution, this means $\Omega\left(\delta_f/p\varepsilon^2 + \delta_g/\varepsilon^2\right)$ and implies $\Omega\left(\delta_f/\varepsilon^2 + p\delta_g/\varepsilon^2\right)$, $\Omega\left(\delta_f/p\varepsilon^2 + \delta_g/\varepsilon^2\right)$ evaluations of $\nabla f$, $\nabla g$, respectively. One can note that the oracle complexity of $\nabla f$ calls can not be better than $\Omega\left(\delta_f/\varepsilon^2\right)$. Thus, values of $p$ that are less than $p = \delta_f/\delta_g$ have no sense. At the same time, $p \geq \delta_f/\delta_g$ makes the estimate on $\nabla g$ calls worse than $\Omega\left(\delta_g/\varepsilon^2\right)$. Thus, the choice of $p = \delta_f/\delta_g$ leads to optimal rates for both oracles simultaneously. $\qquad\square$

## F  PROOF OF COROLLARY 2

Before proceeding to the proof of Corollary 2, we first introduce omitted lemma and theorem.

**Lemma 3.** *Suppose Assumptions 1, 3 hold. Then, for Algorithm 1 it implies*

$$\mathbb{E}_{e^{t+1}}\left[\|e^{t+1} - \nabla(h - h_1)(x^{t+1}, y^{t+1})\|^2\right] \leq \left(1 - \frac{p}{2}\right)\|e^t - \nabla(h - h_1)(x^t, y^t)\|^2$$

$$+ \frac{12}{p}\delta_f^2 \|x^{t+1} - x^t\|^2$$

$$+ \frac{12}{p}\delta_{xy}^2 \|y^{t+1} - y^t\|^2.$$

*Proof.* Firstly, note that

$$\nabla(h - h_1)(x_1, y_1) - \nabla(h - h_1)(x_2, y_2)$$

$$= \int_0^1 \nabla^2(h - h_1)(x_2 + \tau(x_1 - x_2), y_2 + \tau(y_1 - y_2)) \cdot \{x_1 - x_2, y_1 - y_2\} \mathrm{d}\tau.$$

Further, we isolate partial derivatives in $x$:

$$\nabla_x(h - h_1)(x_1, y_1) - \nabla_x(h - h_1)(x_2, y_2)$$

$$= \int_0^1 \nabla_{xx}^2(h - h_1)(x_2 + \tau(x_1 - x_2), y_2 + \tau(y_1 - y_2)) \cdot (x_1 - x_2) \mathrm{d}\tau$$

$$+ \int_0^1 \nabla_{xy}^2(h - h_1)(x_2 + \tau(x_1 - x_2), y_2 + \tau(y_1 - y_2)) \cdot (y_1 - y_2) \mathrm{d}\tau.$$

Applying Assumption 3 to this equality, we obtain

$$\|\nabla_x(h - h_1)(x_1, y_1) - \nabla_x(h - h_1)(x_2, y_2)\| \leq \delta_x \|x_1 - x_2\| + \delta_{xy}\|y_1 - y_2\|.$$

Now let us move to the main part of proof. We start writing out the drift term same as in Lemma 1. We get

$$\mathbb{E}_{e^{t+1}}\left[\|e^{t+1} - \nabla(h - h_1)(x^{t+1}, y^{t+1})\|^2\right]$$

$$= (1 - p)\|e^t + \{0, \nabla_y(h - h_1)(x^{t+1}, y^{t+1})\} - \{0, \nabla_y(h - h_1)(x^t, y^t)\} - \nabla(h - h_1)(x^{t+1}, y^{t+1})\|^2.$$

Next, we add and subtract $\nabla(h - h_1)(x^t, y^t)$ in combination with Young's inequality ($c = p/2$) and obtain

$$E_{e^{t+1}}\left[\|e^{t+1} - \nabla(h - h_1)(x^{t+1}, y^{t+1})\|^2\right]$$

$$\leq \left(1 - \frac{p}{2}\right)\|e^t - \nabla(h - h_1)(x^t, y^t)\|^2 + \frac{2}{p}\|\nabla_x(h - h_1)(x^{t+1}, y^{t+1}) - \nabla_x(h - h_1)(x^t, y^t)\|^2$$

$$\leq \left(1 - \frac{p}{2}\right)\|e^t - \nabla(h - h_1)(x^t, y^t)\|^2 + \frac{4}{p}\delta_x^2\|x^{t+1} - x^t\|^2 + \frac{4}{p}\delta_{xy}^2\|y^{t+1} - y^t\|^2.$$

This concludes the proof. $\qquad\square$

**Theorem 6.** *Suppose Assumptions 1, 2 hold. Consider* $\theta \leq \min\left\{1/8(\delta_x+\delta_y+2\delta_{xy}), p/16\sqrt{2}\max\{\delta_x, \delta_{xy}\}\right\}$. *Then, Algorithm 1 requires*

$$\mathcal{O}\left(\frac{\delta_x + \delta_y + \delta_{xy}}{\varepsilon^2} + \frac{\delta_x + \delta_{xy}}{p\varepsilon^2}\right) \text{ iterations}$$

*to achieve an arbitrary $\varepsilon$-solution, where* $\varepsilon^2 = \mathbb{E}\left[\left\|\frac{1}{T}\sum_{t=1}^{T}\nabla h\left(x^t, y^t\right)\right\|^2\right]$.

*Proof.* Let us denote $z^t = \{x^t, y^t\}$. Same as in Theorem 1 we derive the analogue of smoothness-based descent lemma. Let us start with

$$h(z^{t+1}) - h(z^t) = \int_0^1 \mathrm{d}h(z^t + \tau(z^{t+1} - z^t))\mathrm{d}\tau$$

$$= \int_0^1 \langle \nabla h(z^t + \tau(z^{t+1} - z^t)), z^{t+1} - z^t \rangle$$

and

$$h_1(z^{t+1}) - h_1(z^t) = \int_0^1 \mathrm{d}h_1(z^t + \tau(z^{t+1} - z^t))\mathrm{d}\tau = \int_0^1 \langle \nabla h_1(z^t + \tau(z^{t+1} - z^t)), z^{t+1} - z^t \rangle.$$

Summing up this inequalities, we obtain

$$h(z^{t+1}) - h(z^t) = \int_0^1 \langle \nabla(h - h_1)(z^t + \tau(z^{t+1} - z^t)), z^{t+1} - z^t \rangle \mathrm{d}\tau \tag{10}$$
$$+ h_1(z^{t+1}) - h_1(z^t).$$

Since $z^{t+1}$ is the minimum of $B_\theta^t$ defined in Line 3, we have

$$h_1(z^{t+1}) - h(z^t) \leq -\langle e^t, z^{t+1} - z^t \rangle - \frac{1}{2\theta}\|z^{t+1} - z^t\|^2.$$

Substituting it into equation 10 and applying $\nabla(h - h_1)(z^t)$ as a smart zero, we get

$$h(z^{t+1}) - h(z^t) \leq -\frac{1}{2\theta}\|z^{t+1} - z^t\|^2$$
$$+ \int_0^1 \langle \nabla(h - h_1)(z^t + \tau(z^{t+1} - z^t)) - \nabla(h - h_1)(z^t), z^{t+1} - z^t \rangle \mathrm{d}\tau$$
$$+ \langle \nabla(h - h_1)(z^t) - e^t, z^{t+1} - z^t \rangle.$$

After applying Young's inequality, this turns into

$$h(z^{t+1}) - h(z^t) \leq -\frac{1}{2\theta}\|z^{t+1} - z^t\|^2$$
$$+ \int_0^1 \langle \nabla(h - h_1)(z^t + \tau(z^{t+1} - z^t)) - \nabla(h - h_1)(z^t), z^{t+1} - z^t \rangle \mathrm{d}\tau \tag{11}$$
$$+ \frac{\alpha}{2}\|\nabla(h - h_1)(z^t) - e^t\|^2 + \frac{1}{2\alpha}\|z^{t+1} - z^t\|^2.$$

Let us consider the integral separately. We have

$$\int_0^1 \langle \nabla(h - h_1)(z^t + \tau(z^{t+1} - z^t)) - \nabla(h - h_1)(z^t), z^{t+1} - z^t \rangle \mathrm{d}\tau$$

$$\leq \int_0^1 \|\nabla(h - h_1)(z^t + \tau(z^{t+1} - z^t)) - \nabla(h - h_1)(z^t)\|\|z^{t+1} - z^t\|\mathrm{d}\tau$$

$$\leq \int_0^1 (\delta_x + \delta_y + 2\delta_{xy})\|z^t + \tau(z^{t+1} - z^t) - z^t\|\|z^{t+1} - z^t\|\mathrm{d}\tau$$

$$= \int_0^1 \tau(\delta_x + \delta_y + 2\delta_{xy})\|z^{t+1} - z^t\|^2\mathrm{d}\tau = (\delta_x + \delta_y + 2\delta_{xy})\|z^{t+1} - z^t\|^2.$$

We substitute this into equation 11 with $\alpha = 2\theta$ and derive

$$h(z^{t+1}) - h(z^t) \leq \left(-\frac{1}{4\theta} + \delta_x + \delta_y + 2\delta_{xy}\right)\|z^{t+1} - z^t\|^2 + \theta\|e^t - \nabla(h - h_1)(z^t)\|^2. \tag{12}$$

Let us consider $\Phi^t = h(z^t) - h(z^*) + A\|e^t - \nabla(h - h_1)(z^t)\|^2$ as a potential function. We begin with writing a recursion

$$\Phi^{t+1} = [h(z^t) - h(z^*)] + [h(z^{t+1}) - h(z^t)] + A\|e^{t+1} - \nabla(h - h_1)(z^{t+1})\|^2$$

$$\leq [h(z^t) - h(z^*)] + \theta\|e^t - \nabla(h - h_1)(z^t)\|^2 + \left(-\frac{1}{4\theta} + \delta_x + \delta_y + 2\delta_{xy}\right)\|z^{t+1} - z^t\|^2$$

$$+ A\|e^{t+1} - \nabla(h - h_1)(z^{t+1})\|^2,$$

where the last transition exploits equation 12. Next, we apply Lemma 3 and derive

$$\mathbb{E}_{e^{t+1}}\left[\Phi^{t+1}\right] = [h(z^t) - h(z^*)] + [h(z^{t+1}) - h(z^t)] + A\mathbb{E}_{e^{t+1}}\left[\|e^{t+1} - \nabla(h - h_1)(z^{t+1})\|^2\right]$$

$$\leq [h(z^t) - h(z^*)] + \theta\|e^t - \nabla(h - h_1)(z^t)\|^2$$

$$+ \left(-\frac{1}{4\theta} + \delta_x + \delta_y + 2\delta_{xy}\right)\|z^{t+1} - z^t\|^2 + A\left(1 - \frac{p}{2}\right)\|e^t - \nabla(h - h_1)(z^t)\|$$

$$+ \frac{4A}{p}\delta_x^2\|x^{t+1} - x^t\|^2 + \frac{4A}{p}\delta_{xy}^2\|y^{t+1} - y^t\|^2$$

$$\leq [h(z^t) - h(z^*)] + \theta\|e^t - \nabla(h - h_1)(z^t)\|^2$$

$$+ \left(-\frac{1}{4\theta} + \delta_x + \delta_y + 2\delta_{xy}\right)\|z^{t+1} - z^t\|^2 + \frac{8A}{p}\max\{\delta_x^2, \delta_{xy}^2\}\|z^{t+1} - z^t\|^2.$$

After grouping terms, we arrive at

$$\mathbb{E}_{e^{t+1}}\left[\Phi^{t+1}\right] \leq \Phi^t + \left(-\frac{1}{4\theta} + \delta_x + \delta_y + 2\delta_{xy} + \frac{8A}{p}\max\{\delta_x^2, \delta_{xy}^2\}\right)\|z^{t+1} - z^t\|^2$$

$$+ \left(\theta - \frac{Ap}{2}\right)\|e^t - \nabla(h - h_1)(z^t)\|. \tag{13}$$

Let us deal with the first term of equation 13. First, let us note that Line 3 of Algorithm 2 implies

$$0 = e^t + \nabla(h_1 - h)(z^{t+1}) + \nabla h(z^{t+1}) + \frac{z^{t+1} - z^t}{\theta}$$

$$= [e^t - \nabla(h - h_1)(z^t)] + [\nabla(h - h_1)(z^{t+1}) - \nabla(h - h_1)(z^t)] + \nabla h(z^{t+1}) + \frac{z^{t+1} - z^t}{\theta}.$$

This implies

$$\|z^{t+1} - z^t\|^2 = \theta^2\|[e^t - \nabla(h - h_1)(z^t)] + [\nabla(h - h_1)(z^{t+1}) - \nabla(h - h_1)(z^t)] + \nabla h(z^{t+1})\|^2.$$

Next, we employ the trick. We write

$$\|\nabla h(z^{t+1})\|^2 = \|\nabla h(z^{t+1}) + [e^t - \nabla(h - h_1)(z^t)] + [\nabla(h - h_1)(z^{t+1}) - \nabla(h - h_1)(z^t)]$$

$$- [e^t - \nabla(h - h_1)(z^t)] - [\nabla(h - h_1)(z^{t+1}) - \nabla(h - h_1)(z^t)]\|^2$$

$$\leq 3\|\nabla h(z^{t+1}) + [e^t - \nabla(h - h_1)(z^t)] + [\nabla(h - h_1)(z^{t+1}) - \nabla(h - h_1)(z^t)]\|^2$$

$$+ 3\|e^t - \nabla(h - h_1)(z^t)\|^2 + 3\|\nabla(h - h_1)(z^{t+1}) - \nabla(h - h_1)(z^t)\|^2.$$

This implies

$$\|\nabla h(z^{t+1}) + [e^t - \nabla(h - h_1)(z^t)] + [\nabla(h - h_1)(z^{t+1}) - \nabla(h - h_1)(z^t)]\|^2$$

$$\geq \frac{1}{3}\|\nabla h(z^{t+1})\|^2 - \|e^t - \nabla(h - h_1)(z^t)\|^2 - \|\nabla(h - h_1)(z^{t+1}) - \nabla(h - h_1)(z^t)\|^2$$

$$\geq \frac{1}{3}\|\nabla h(z^{t+1})\|^2 - \|e^t - \nabla(h - h_1)(z^t)\|^2 - (\delta_x + \delta_y + 2\delta_{xy})^2\|z^{t+1} - z^t\|^2.$$

Thus, there is a lower estimate on the update:

$$\|z^{t+1} - z^t\|^2 \geq \frac{\theta^2}{3}\|\nabla h(z^{t+1})\|^2 - \theta^2\|e^t - \nabla(h - h_1)(z^t)\|^2$$

$$- \theta^2(\delta_x + \delta_y + 2\delta_{xy})^2\|z^{t+1} - z^t\|^2.$$

After rearranging terms, we get

$$\left(1 - \theta^2(\delta_x + \delta_y + 2\delta_{xy})^2\right)\|z^{t+1} - z^t\|^2 \geq \frac{\theta^2}{3}\|\nabla h(z^{t+1})\|^2 - \theta^2\|e^t - \nabla(h - h_1)(z^t)\|^2.$$

Due to the choice of parameters outlined in Theorem 6, we have $\theta \leq 1/2(\delta_x + \delta_y + 2\delta_{xy})$ which implies

$$1 - \theta^2(\delta_x + \delta_y + 3\delta_{xy})^2 \geq 1 - \frac{1}{4} = \frac{3}{4}.$$

Thus, we have

$$\|z^{t+1} - z^t\|^2 \geq \frac{4\theta^2}{9}\|\nabla h(z^{t+1})\|^2 - \frac{4\theta^2}{3}\|e^t - \nabla(h - h_1)(z^t)\|^2. \tag{14}$$

Comparing equation 13 and equation 14, we observe that the variance could be included in the resulting inequality with negative sign only if $A < 2\theta/p$. We choose $A = 4\theta/p$. This means that equation 13 transforms into

$$\mathbb{E}_{e^{t+1}}\left[\Phi^{t+1}\right] \leq \Phi^t + \left(-\frac{1}{4\theta} + \delta_x + \delta_y + 2\delta_{xy} + \frac{32\theta \max\{\delta_x^2, \delta_{xy}^2\}}{p^2}\right)\|z^{t+1} - z^t\|^2$$
$$- \theta\|e^t - \nabla(h - h_1)(z^t)\|.$$

To substitute equation 14, the first term of this inequality should be negative. Let us show that it is actually negative because of choice of $\theta$ (see Theorem 6). On the one hand, $\theta \leq 1/8(\delta_x + \delta_y + 2\delta_{xy})$, which implies $(\delta_x + \delta_y + 2\delta_{xy}) \leq 1/8\theta$. On the other hand, $\theta \leq p/16\sqrt{2}\max\{\delta_x, \delta_{xy}\}$, which implies $32\theta \max\{\delta_x^2, \delta_{xy}^2\}/p^2 \leq 1/16\theta$. Thus, we have

$$-\frac{1}{4\theta} + \delta_x + \delta_y + 2\delta_{xy} + \frac{32\theta \max\{\delta_x^2, \delta_{xy}^2\}}{p^2} \leq -\frac{1}{16\theta} < 0.$$

Thus, we can apply equation 14 and deduce

$$\mathbb{E}_{e^{t+1}}\left[\Phi^{t+1}\right] \leq \Phi^t - \frac{\theta}{36}\|\nabla h(z^{t+1})\|^2 + \left(\frac{1}{12} - 1\right)\theta\|e^t - \nabla(h - h_1)(z^t)\|^2,$$

which implies

$$\mathbb{E}_{e^{t+1}}\left[\|\nabla h(z^{t+1})\|^2\right] \leq \frac{36}{\theta}[\Phi^t - \Phi^{t+1}].$$

Taking full expectation and accounting for $e^0 = \nabla(h - h_1)(z^0)$, we obtain

$$\mathbb{E}\left[\left\|\frac{1}{T}\sum_{t=1}^{T}\nabla h(z^t)\right\|^2\right] \leq \frac{36[h(z^0) - h(z^*)]}{\theta}$$

$$= \mathcal{O}\left(\left(\frac{\delta_x + \delta_y + \delta_{xy}}{T} + \frac{\delta_x + \delta_{xy}}{pT}\right)[h(z^0) - h(z^*)]\right).$$

This concludes the proof of Theorem 6. $\qquad\square$

Now we are ready to move on to the corollary.

**Corollary 4** (Corollary 2). *Suppose Assumptions 1, 3. Consider $\theta \leq \min\left\{1/8(\delta_x + \delta_y + 2\delta_{xy}), p/16\sqrt{2}\max\{\delta_x, \delta_{xy}\}\right\}$. Algorithm 2 with $p = (\delta_x + \delta_{xy})/(\delta_x + \delta_y + \delta_{xy})$ requires*

$$\mathcal{O}\left(\frac{d_x\delta_x}{\varepsilon^2} + \frac{d_y\delta_y}{\varepsilon^2} + \frac{(d_x + d_y)\delta_{xy}}{\varepsilon^2}\right) \text{ bits}$$

*to reach an arbitrary $\varepsilon$-solution.*

*Proof.* Theorem 6 implies that $\nabla_x h$ is called $\mathcal{O}\left(p(\delta_x + \delta_y + \delta_{xy})/\varepsilon^2 + (\delta_x + \delta_{xy})/\varepsilon^2\right)$ times and $\nabla_y h$ is evaluated $\mathcal{O}\left((\delta_x + \delta_y + \delta_{xy})/\varepsilon^2 + (\delta_x + \delta_{xy})/p\varepsilon^2\right)$. The choice of $p$ results in $\mathcal{O}\left((\delta_x + \delta_{xy})/\varepsilon^2\right)$ and $\mathcal{O}\left((\delta_y + \delta_{xy})/\varepsilon^2\right)$. Every evaluation of $\nabla_x h$ requires communicating $d_x$ units of information, and $\nabla_y h$ requires $d_y$. This implies the result of Corollary 2. $\qquad\square$

# G  PROOF OF THEOREM 3

**Theorem 7.** *(Theorem 3) There exists such $h$, satisfying Assumptions 1, 2, that any algorithm $\mathcal{A}$ (see Definition 1) requires to transmit*

$$\Omega\left(\frac{d_x\delta_x}{\varepsilon^2} + \frac{d_y\delta_y}{\varepsilon^2} + \frac{(d_x + d_y)\delta_{xy}}{\varepsilon^2}\right) \text{ bits}$$

*to reach an arbitrary $\varepsilon$-solution when $\delta_{xy} < \delta_x$.*

*Proof.* The proof of this theorem is equivalent to the proof of Theorem 2. By considering the same function as in Theorem 2, where function $f$ depends only on $x$, and function $g$ depends only on $y$, we obtain that the effective $\hat{\delta}_{xy}$ (i.e., the minimal possible one) equals zero. This implies that we obtain lower bounds on the oracle complexity:

$$\Omega\left(\frac{\delta_x}{\varepsilon^2}\right) \text{ and } \Omega\left(\frac{\delta_y}{\varepsilon^2}\right)$$

oracle calls. Moreover, in the case $\delta_{xy} < \delta_x < \delta_y$, we have that

$$\Omega\left(\frac{\delta_x + \delta_{xy}}{\varepsilon^2}\right) = \Omega\left(\frac{\delta_x}{\varepsilon^2}\right)$$

and

$$\Omega\left(\frac{\delta_y + \delta_{xy}}{\varepsilon^2}\right) = \Omega\left(\frac{\delta_y}{\varepsilon^2}\right).$$

Therefore, we get that the lower bound on bit complexities for each block of coordinates can be expressed as

$$\Omega\left(\frac{d_x(\delta_x + \delta_{xy})}{\varepsilon^2}\right) \text{ and } \Omega\left(\frac{d_y(\delta_y + \delta_{xy})}{\varepsilon^2}\right).$$

Summing up, we obtain the final bound. □

## H   INEXACT INNER MINIMIZATION

Algorithms 1, 2 require access to Proximal Incremental First-Order Oracle defined in Section 6.1.1. This is impractical, because the minimization of $A_{\theta^t}$ and $B_{\theta^t}$ cannot be performed in negligible time, since $h_1$ is not merely a regularizer but an empirical risk constructed from the model's output. Nevertheless, in our experiments we observe that performing three epochs of `Adam` on the server's side are sufficient to obtain a stable optimization trajectory despite the inexact solution of the sub-problem. Below, we present the results of a numerical study of robustness to inexact minimization.

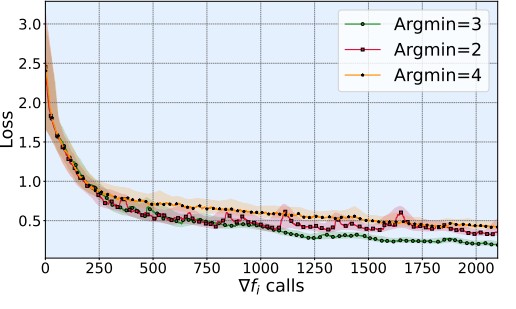 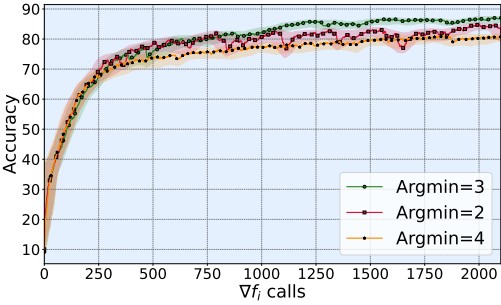

Figure 7: Dynamics of `HASCA` with various number of epochs to solve the inner minimization. `Adam` is used as an optimizer.

Figure 7 shows the existence of an optimal number of `Adam` epochs for solving the inner minimization. We attribute this to the fact that too few epochs yield an overly inaccurate solution, whereas an excessively large number leads to overfitting to the server's data.

To investigate the observed phenomenon, we introduce an additional assumption on $h_1$.

**Assumption 4.** *The function $h_1$ is convex, i.e. for every $x, y \in \mathbb{R}^d$*

$$h_1(x) - h_1(y) \geq \langle \nabla h_1(y), x - y \rangle.$$

Although neural networks are inherently non-convex, theoretical analysis under convexity of the server remains relevant. Recent studies suggest that deep neural networks often exhibit properties similar to convexity in certain regions, making insights from convex analysis applicable (Kleinberg et al., 2018; Zhou et al., 2019; Liu et al., 2022). Moreover, convex optimization serves as a theoretical foundation for the design of optimization algorithms.

Now, let us prove the convergence of Algorithm 1 with inexact inner minimzation.

**Theorem 8.** *Suppose Assumptions 1, 2, 4 hold. Consider $\theta \leq \min\{1/8(\delta_f+\delta_g), p/8\sqrt{2}\delta_f\}$. Let the subproblem in Line 3 be solved with precision*

$$\|\nabla A_\theta^t(x^{t+1})\|^2 \leq \frac{3}{14\theta^2}\|x^t - \arg\min A_\theta^t(x)\|^2.$$

*Then, Algorithm 1 requires*

$$\mathcal{O}\left(\frac{\delta_f+\delta_g}{\varepsilon^2} + \frac{\delta_f}{p\varepsilon^2}\right) \text{ iterations}$$

*to achieve an arbitrary $\varepsilon$-solution, where $\varepsilon^2 = \mathbb{E}\left[\left\|\frac{1}{T}\sum_{t=1}^T \nabla h(x^t)\right\|^2\right]$.*

*Proof.* Note that the inequality $A_\theta^t(x^{t+1}) \leq A_\theta^t(x^t)$ holds even when the inner minimization is solved with a large error. In fact, a single gradient descent step is sufficient to guarantee its satisfaction. Therefore, the proof of Theorem 1 remains unchanged up to the point where we obtain

$$\mathbb{E}_{e^{t+1}}\left[\Phi^{t+1}\right] \leq \Phi^t + \left(-\frac{1}{4\theta} + \delta_f + \delta_g + \frac{2A}{p}\delta_f^2\right)\|x^{t+1} - x^t\|^2 \tag{15}$$
$$+ \left(\theta - \frac{Ap}{2}\right)\|e^t - \nabla(h-h_1)(x^t)\|.$$

Let us note that Line 3 of Algorithm 1 implies

$$\nabla A_\theta^t(x^{t+1}) = e^t + \nabla(h_1 - h)(x^{t+1}) + \nabla h(x^{t+1}) + \frac{x^{t+1} - x^t}{\theta}$$
$$= [e^t - \nabla(h-h_1)(x^t)] + [\nabla(h-h_1)(x^{t+1}) - \nabla(h-h_1)(x^t)] + \nabla h(x^{t+1})$$
$$+ \frac{x^{t+1} - x^t}{\theta}.$$

Next, we write

$$\|\nabla h(x^{t+1})\|^2 = \|\nabla h(x^{t+1}) + [e^t - \nabla(h-h_1)(x^t)] + [\nabla(h-h_1)(x^{t+1}) - \nabla(h-h_1)(x^t)]$$
$$- \nabla A_\theta^t(x^{t+1}) - [e^t - \nabla(h-h_1)(x^t)]$$
$$- [\nabla(h-h_1)(x^{t+1}) - \nabla(h-h_1)(x^t)]\|^2$$
$$\leq 4\|\nabla h(x^{t+1}) + [e^t - \nabla(h-h_1)(x^t)] + [\nabla(h-h_1)(x^{t+1}) - \nabla(h-h_1)(x^t)]$$
$$- \nabla A_\theta^t(x^{t+1})\|^2$$
$$+ 4\|e^t - \nabla(h-h_1)(x^t)\|^2 + 4\|\nabla(h-h_1)(x^{t+1}) - \nabla(h-h_1)(x^t)\|^2$$
$$+ 4\|\nabla A_\theta^t(x^{t+1})\|^2.$$

This implies

$$\|\nabla h(x^{t+1}) + [e^t - \nabla(h-h_1)(x^t)] + [\nabla(h-h_1)(x^{t+1}) - \nabla(h-h_1)(x^t)] - \nabla A_\theta^t(x^{t+1})\|^2$$
$$\geq \frac{1}{3}\|\nabla h(x^{t+1})\|^2 - \|e^t - \nabla(h-h_1)(x^t)\|^2 - (\delta_f + \delta_g)^2\|x^{t+1} - x^t\|^2 - \theta^2\|\nabla A_\theta^t(x^{t+1})\|^2.$$

Thus, there is a lower bound on the update:

$$\|x^{t+1} - x^t\|^2 \geq \frac{\theta^2}{4}\|\nabla h(x^{t+1})\|^2 - \theta^2\|e^t - \nabla(h-h_1)(x^t)\|^2 - \theta^2(\delta_f + \delta_g)^2\|x^{t+1} - x^t\|^2$$
$$- \theta^2\|\nabla A_\theta^t(x^{t+1})\|^2.$$

After rearranging terms, we get

$$\left(1 - \theta^2(\delta_f + \delta_g)^2\right)\|x^{t+1} - x^t\|^2 \geq \frac{\theta^2}{4}\|\nabla h(x^{t+1})\|^2 - \theta^2\|e^t - \nabla(h-h_1)(x^t)\|^2$$
$$- \theta^2\|\nabla A_\theta^t(x^{t+1})\|^2.$$

Due to the choice of parameters outlined in Theorem 8, we have $\theta \leq 1/2(\delta_f+\delta_g)$ which implies

$$1 - \theta^2(\delta_f + \delta_g)^2 \geq 1 - \frac{1}{4} = \frac{3}{4}.$$

Thus, we have

$$\|x^{t+1} - x^t\|^2 \ge \frac{\theta^2}{3}\|\nabla h(x^{t+1})\|^2 - \frac{4\theta^2}{3}\|e^t - \nabla(h - h_1)(x^t)\|^2 - \frac{4\theta^2}{3}\|\nabla A_\theta^t(x^{t+1})\|^2. \quad (16)$$

Comparing equation 15 and equation 16, we observe that the variance could be included in the resulting inequality with negative sign only if $A > 2\theta/p$. We choose $A = 4\theta/p$. This means that equation 8 transforms into

$$\mathbb{E}_{e^{t+1}}\left[\Phi^{t+1}\right] \le \Phi^t + \left(-\frac{1}{4\theta} + \delta_f + \delta_g + \frac{8\theta\delta_f^2}{p^2}\right)\|x^{t+1} - x^t\|^2 - \theta\|e^t - \nabla(h - h_1)(x^t)\|.$$

To substitute equation 9, the first term of this inequality should be negative. Let us show that it is actually negative because of choice of $\theta$ (see Theorem 8). On the one hand, $\theta \le 1/8(\delta_f+\delta_g)$, which implies $(\delta_f + \delta_g) \le 1/8\theta$. On the other hand, $\theta \le p/8\sqrt{2}\delta_f$, which implies $8\theta\delta_f^2/p^2 \le 1/16\theta$. Thus, we have

$$-\frac{1}{4\theta} + \delta_f + \delta_g + \frac{8\theta\delta_f^2}{p^2} \le -\frac{1}{16\theta} < 0,$$

which implies

$$\mathbb{E}_{e^{t+1}}\left[\Phi^{t+1}\right] \le \Phi^t - \frac{1}{32\theta}\|x^{t+1} - x^t\|^2 - \frac{1}{32\theta}\|x^{t+1} - x^t\|^2 - \theta\|e^t - \nabla(h - h_1)(x^t)\|.$$

We can apply equation 9 and deduce

$$\mathbb{E}_{e^{t+1}}\left[\Phi^{t+1}\right] \le \Phi^t - \frac{\theta}{96}\|\nabla h(x^{t+1})\|^2 + \left(\frac{1}{24} - 1\right)\theta\|e^t - \nabla(h - h_1)(x^t)\|^2 - \frac{1}{32\theta}\|x^{t+1} - x^t\|$$

$$+ \frac{\theta}{24}\|\nabla A_\theta^t(x^{t+1})\|$$

$$\le \Phi^t - \frac{\theta}{96}\|\nabla h(x^{t+1})\|^2 - \frac{1}{32\theta}\|x^{t+1} - x^t\| + \frac{\theta}{24}\|\nabla A_\theta^t(x^{t+1})\|^2.$$

Next, we consider the last two terms separately.

$$-\frac{1}{32\theta}\|x^{t+1} - x^t\|^2 + \frac{\theta}{24}\|\nabla A(x^{t+1})\|^2 \le -\frac{1}{64\theta}\|x^t - \arg\min A_\theta^t(x)\|^2$$

$$+ \frac{1}{32\theta}\|x^{t+1} - \arg\min A_\theta^t(x)\|^2 + \frac{\theta}{24}\|\nabla A_\theta^t(x)\|^2.$$

Let us take into account the $1/\theta$-strong convexity of $A_\theta^t$ and write

$$-\frac{1}{32\theta}\|x^{t+1} - x^t\|^2 + \frac{\theta}{24}\|\nabla A(x^{t+1})\|^2 \le -\frac{1}{64\theta}\|x^t - \arg\min A_\theta^t(x)\|^2 + \frac{7\theta}{96}\|\nabla A_\theta^t(x^{t+1})\|^2$$

$$= \frac{7\theta}{96}\left[\|\nabla A_\theta^t(x^{t+1})\|^2 - \frac{3}{14\theta^2}\|x^t - \arg\min A_\theta^t(x)\|^2\right].$$

Using precision of the inner minimization (Theorem 8), we obtain

$$\mathbb{E}_{e^{t+1}}\left[\|\nabla h(x^{t+1})\|^2\right] \le \frac{36}{\theta}[\Phi^t - \Phi^{t+1}].$$

Taking full expectation and accounting for $e^0 = \nabla(h - h_1)(x^0)$, we obtain

$$\mathbb{E}\left[\left\|\frac{1}{T}\sum_{t=1}^T \nabla h(x^t)\right\|^2\right] \le \frac{96[h(x^0) - h(x^*)]}{\theta} = \mathcal{O}\left(\left(\frac{\delta_f + \delta_g}{T} + \frac{\delta_f}{pT}\right)[h(x^0) - h(x^*)]\right).$$

This concludes the proof of Theorem 8. $\qquad\square$

Since the proof in the coordinate setup is conceptually analogous, the result for inexact minimization in Algorithm 2 can be obtained analogously.

## THE USE OF LARGE LANGUAGE MODELS (LLMS)

Language models were used to improve text quality (mostly to correct grammatical errors). LLMs were not used to obtain theoretical results or write code.

