# OpenReview forum: "Complexity-Separated Schemes for Addressing Structured Heterogeneity in Federated Learning"
_ICLR.cc/2026/Conference — Submitted to ICLR 2026_

### Official Review · Reviewer_Ht2T · 2025-10-27

**Soundness:** 3
**Presentation:** 3
**Contribution:** 3
**Rating:** 6
**Confidence:** 3

**Summary:**

The paper introduces a well-motivated approach to federated optimization by exploiting structured heterogeneity, with rigorous theoretical foundations and empirical validation. The algorithms demonstrate clear communication efficiency gains in non-convex settings, as shown in theoretical bounds (Theorems 1–3, Corollaries 1–2) and experiments (Figures 1–2, Table 1). However, the experimental section lacks detailed ablation studies and reproducibility information, such as hyperparameter tuning and statistical significance tests. Additionally, comparisons with closely related methods (e.g., ProxyProx) could be more explicit in both theory and experiments.

**Strengths:**

The paper is the first to systematically address structured heterogeneity in non-convex federated learning, separating complexity for distribution-based and coordinate-based cases.

The analysis includes upper bounds, matching lower bounds, and detailed proofs leveraging zero-chain functions and Hessian similarity assumptions.

**Weaknesses:**

Only two datasets (CIFAR-10, Avito) and two neural architectures (ResNet-18, BERT) are used, raising questions about generalizability.

Differences from ProxyProx and Accelerated ExtraGradient are not thoroughly analyzed in terms of convergence speed or robustness.

Theoretical results depend on Hessian similarity, but no empirical validation of these assumptions is provided.

The impact of algorithmic components (e.g., probability pp, reference points) is not systematically ablated.

Experimental results lack error bars or repeated runs, making it difficult to assess robustness.

Scenarios where HASCA or C-HASCA underperform are not discussed, such as extreme heterogeneity or noisy clients. No direct evidence found in the manuscript.

The subproblem in Line 3 of Algorithms 1–2 requires exact minimization, but no analysis of its computational cost is provided.

**Questions:**

Please respond to the Weaknesses.

---

> ### Author Response · Authors · 2025-11-21
>
> Dear Reviewer Ht2T!
>
> Thank you for positive evaluation of our work and for suggestions on further improvements!
>
> >Only two datasets (CIFAR-10, Avito) and two neural architectures (ResNet-18, BERT) are used, raising questions about generalizability.
>
> We agree that extending our experimental section to more complex problems would strengthen our work. We address the problem of fine-tuning FasterViT 270M for Food101 classification. The table below shows the dynamics of test accuracy.
> |# of $\nabla f$ calls|500|1000|1500|2000|2250|
> |-|-|-|-|-|-|
> |Adam|46.53|62.46|68.72|73.90|74.43|
> |HASCA|47.40|63.74|71.26|74.85|75.40|
>
> We have added this experiment to Appendix A6.
>
> >Differences from ProxyProx and Accelerated ExtraGradient are not thoroughly analyzed in terms of convergence speed or robustness.
>
> We have addressed this issue in Appendix A5 of the revised PDF. Let us consider the setup from Section 6.1. Accelerated ExtraGradient and ProxyProx have the same number of $\nabla f$ and $\nabla g$ calls that directly depends on $\delta_f + \delta_g$. In contrast, for our method, the number of $\nabla f$ and $\nabla g$ calls depends only on $\delta_f$ and $\delta_g$, respectively. As $\kappa \to 1$, $\delta_f$ decreases while $\delta_f + \delta_g$ increases. This implies that our method requires fewer $\nabla f$ calls, whereas the complexity of the competing methods increases. This effect is observed in Tables 3, 4 for $\kappa = \{0.8,0.95}$. Conversely, when $\kappa \to 0.5$, $\delta_f$ increases while $\delta_f + \delta_g$ decreases, and differences between our scheme and competitors become less pronounced. This effect is shown in Table 2. When $\kappa = 0.5$, $\delta_f$ and $\delta_g$ are equal, and HASCA becomes equivalent to ProxyProx. The same set of reasoning holds for Section 6.2.
>
> >Theoretical results depend on Hessian similarity, but no empirical validation of these assumptions is provided.
>
> In our work, we assume Hessian similarity for separately available oracles to consider an FL-motivated problem with multiple heterogeneity constants. Since Hessian similarity is well-established in the optimization literature [1, 2, 3], we do not empirically validate it and concentrate on developing the theory for FL settings.
>
> >The impact of algorithmic components (e.g., probability pp, reference points) is not systematically ablated.
>
> Below, we present a study of the effect of $p$ on the performance of Algorithm 1 for $\kappa = 0.8$. We are ready to provide results for other values of $\kappa$ if necessary.
> |p|Test accuracy, $\%$|
> |-|-|
> |0.3|84.38|
> |0.4|83.04|
> |0.5|86.82|
> |0.6|81.73|
> |0.7|81.06|
>
> We have added figures and a discussion of the observed effects in Appendix A3. We also highlight that reference points are not hyperparameters and are determined based on the optimization trajectory of the algorithm.
>
> >Experimental results lack error bars or repeated runs, making it difficult to assess robustness.
>
> We are not entirely clear about Reviewer’s concern, as all plots in our work include error bars.
>
> >Scenarios where HASCA or C-HASCA underperform are not discussed, such as extreme heterogeneity or noisy clients. No direct evidence found in the manuscript.
>
> The response to this question was provided in our second comment. The algorithms we proposed underperform as $\kappa \to 0.5$, i.e., when the server sample equally well covers all modes of the data distribution. A detailed discussion can be found in Appendix A5.
>
> >The subproblem in Line 3 of Algorithms 1–2 requires exact minimization, but no analysis of its computational cost is provided.
>
> In our work, three local Adam epochs on the server side are sufficient for good convergence of the method. However, ablations show that the method still converges with fewer epochs.
> |Epochs|Final test accuracy|
> |-|-|
> |2|83.69|
> |3|86.82|
> |4|80.5|
>
> Moreover, we have modified our proofs to account for inexact minimization. We have derived a precision criterion for inner minimization that must be satisfied to ensure convergence. For analysis and experiments, see Appendix H.
>
> We are ready to continue the discussion if Reviewer has any remaining concerns. If we have addressed all questions satisfactorily, we would kindly ask them to reconsider the score.
>
> ---
>
> **References:**
>
> [1] Hendrikx, H., et al. “Statistically preconditioned accelerated gradient method for distributed optimization”. **ICML-20**
>
> [2] Kovalev, D., et al. “Optimal Gradient Sliding and its Application to Distributed Optimization Under Similarity”. **NeurIPS-22**
>
> [3] Takezawa, Y., et al. “Exploiting Similarity for Computation and Communication-Efficient Decentralized Optimization”. **ICML-25**

---

> ### Author Response · Authors · 2025-11-27
>
> Dear Reviewer Ht2T!
>
> We respectfully follow up on our rebuttal, as we have not yet received your response. We fully recognize that author–reviewer discussion can be time-consuming, and we greatly value the effort and attention you devote to evaluating submissions. If you have the opportunity, we would be grateful to receive your feedback on our reply and to engage in further discussion.
>
> We sincerely appreciate your time and careful consideration!

---

### Official Review · Reviewer_witW · 2025-10-29

**Soundness:** 1
**Presentation:** 1
**Contribution:** 1
**Rating:** 0
**Confidence:** 5

**Summary:**

The paper's goal is to decouple two forms of data heterogeneity in federated learning and design communication-efficient optimization algorithms. The algorithm is similar to FedProx, but applied to composite functions. The analysis and experimental results are quite rudimentary and limited.

**Strengths:**

The goal of the paper is to consider two different forms of data hetergeneity: mode-based and coordinate-based.

**Weaknesses:**

I have several major concerns about this paper:

* The paper spends several pages on motivating and setting up the problem, first talking about distributed learning with IID data splits, then federated learning with data heterogeneity. For an established research topic like federated learning this extended introduction is not necessary. The space would have been better used to present the paper's own algorithm, analysis, and results.

* The notation section talks about 3 measures of communication complexity: 1) communication rounds, which is used in synchronous methods, 2) asynchronous aggregation, and 3) compressed communication. Each of these paradigms pose separate challenges for algorithm design and analyses that have been addressed by a long line of previous works. I don't believe how the authors can unify all of these in their analysis.

* The communication complexity bounds are rudimentary order-wise bounds. Also the order of the rounds in the bounds seems incorrect to me. Without strong convexity assumptions the rate is O(1/\epsilon^2) not O(1/\epsilon).

* Experimental results compare with a limited set of methods. Why haven't the authors considered more standard baselines like FedAvg, FedProx, SCAFFOLD etc? The omission of FedProx, which is closely related to this paper, is particularly glaring. The paper does not even include a citation to FedProx.

**Questions:**

Please clarify what exact problem setting you are working with in this paper. The problem formulation, proposed algorithm and contributions are not clear at all based on my reading of the paper.

**Details Of Ethics Concerns:**

The paper makes some unsubstantiated statements and claims.

---

> ### Author Response · Authors · 2025-11-21
>
> Dear Reviewer witW!
>
> Thank you for taking the time to review our work! Below, we are responding to your claims.
>
> >The paper spends several pages on motivating and setting up the problem, first talking about distributed learning with IID data splits, then federated learning with data heterogeneity. For an established research topic like federated learning this extended introduction is not necessary. The space would have been better used to present the paper's own algorithm, analysis, and results.
>
> In our work, the introduction to distributed/federated learning occupies a single page followed by substantive content. We consider a non-standard FL setting in which data similarity is endowed with a structure: certain modes of the data distribution are more homogeneous compared to others. This poses a novel class of optimization problems that requires a detailed discussion and clear motivation. We also draw Reviewer's attention to the structure of several papers published at top-tier conferences. For example, [1], which received an oral presentation at **NeurIPS-21**, devotes 6 out of 10 pages to preparing the ground for main results. Reviewer can also check other recent works, for instance, orals at **ICLR-25**: [2] (5.5 pages on problem formulation) and [3] (4 pages). We believe that the structure of our paper is fully aligned with norms of A* venues.
>
> >The notation section talks about 3 measures of communication complexity. I don't believe how the authors can unify all of these in their analysis.
>
> We agree that optimal communication complexity cannot be achieved under all definitions simultaneously. Our paper contains no line where we claim otherwise. For the setting of Section 6.1, we provide bounds on the number of $\nabla f$ and $\nabla g$ calls, (communication rounds). Further, we discuss other definitions of complexity (Lines 326-330), without asserting that we achieve optimal rates under those definitions as well. An analogous analysis for $\nabla_x h$ and $\nabla_y h$ is presented in Section 6.2 for Algorithm 2. We refer you to Appendices C, F for verification.
>
> >Without strong convexity assumptions the rate is O(1/\epsilon^2) not O(1/\epsilon).
>
> In the original PDF, we explicitly indicated that we denote $\varepsilon=||\frac{1}{T}\sum_{i=1}^T\nabla h(x^t)||^2$, which is consistent with prior work [5]. With this notation, the rate is $O(1/\varepsilon)$. Works that obtain $O(1/\varepsilon^2)$ use $\varepsilon^2=||\frac{1}{T}\sum_{i=1}^T\nabla h(x^t)||^2$ [4]. We have aligned our notation with one requested by Reviewer.
>
> >Why haven't the authors considered more standard baselines like FedAvg, FedProx, SCAFFOLD etc?
>
> Our work improves the guarantees in the Hessian similarity setting by leveraging the idea of complexity separation. Although the works mentioned by Reviewer employ Hessian similarity in convergence proofs, their complexity depends not only on $\delta$ but also on the smoothness constant $L$ (for example, see Theorem 4 in [9]). For this reason, we did not considered them as baselines. The only exception is Adam, which is included as the most widely used strong baseline. However, to address Reviewer’s concern, we have included FedProx and SCAFFOLD in the experimental section.
>
> >The paper does not even include a citation to FedProx
>
> We do not cite FedProx because it does not fall under either of the two subsections covered in the literature review. Now that it is used as a baseline, we cite it in the experimental section.
>
> >Please clarify what exact problem setting you are working with in this paper.
>
> Methods that leverage data similarity perform well in distributed optimization, where local samples are IID and $\delta \to 0$ as the server dataset size increases. However, they perform poorly in federated learning due to heterogeneity (see Table 3 in [6]). Motivated by experimental observations from prior work, where certain modes of data distribution are more homogeneous across the network compared to others, we introduce structured heterogeneity. Mathematically, this extends the multi-smoothness setting [7,8] to data similarity.
>
> >The paper makes some unsubstantiated statements and claims.
>
> All claims made in our work are supported either by appropriate citations or by formal proofs. We kindly ask you to clarify which specific claim you consider to be unsupported.
>
> We additionally point out that your concerns relate to length of the introduction, absence of the baseline, and misunderstanding of the text and notation (communication complexity, $\varepsilon$). In this context, we believe that current evaluation of the paper is unfair and respectfully request the reconsideration of the score.
>
> ---
>
> For the list of references, see the following comment

---

> ### Author Response · Authors · 2025-11-21
>
> **References:**
>
> [1] Richtarik, P., et al. “EF21: A New, Simpler, Theoretically Better, and Practically Faster Error Feedback”. **NeurIPS-21**
>
> [2] Zhang, J., et al. “AFLOW: AUTOMATING AGENTIC WORKFLOW GENERATION”. **ICLR-25**
>
> [3] Kim, J., Suzuki, T. “TRANSFORMERS PROVABLY SOLVE PARITY EFFICIENTLY WITH CHAIN OF THOUGHT”. **ICLR-25**
>
> [4] Carmon, Y., et al. “Lower Bounds for Finding Stationary Points I”. **Mathematical Programming**, 2020
>
> [5] Ghadimi, S., Lan, G. “STOCHASTIC FIRST- AND ZEROTH-ORDER METHODS
> FOR NONCONVEX STOCHASTIC PROGRAMMING”, **SIAM**, 2013
>
> [6] Karimireddy, SP., et al. “SCAFFOLD: Stochastic Controlled Averaging for Federated Learning”. **ICML-20**
>
> [7] Richtarik, P., Takac’, M. “Iteration complexity of randomized block-coordinate descent methods for minimizing a composite function”. **Mathematical Programming**, 2014
>
> [8] Gladin, E., et al. “Solving smooth min-min and min-max problems by mixed oracle algorithms”, **MOTOR-21**
>
> [9] Luo, R., et al. “Revisiting LocalSGD and SCAFFOLD: Improved Rates and Missing Analysis ”, **AISTATS-25**

---

> > ### Comment · Reviewer_witW · 2025-11-24
> > **Thanks for the clarification**
> >
> > Thanks for the clarification your paper's results, especially about the convergence bound. I will update my review and rating.

---

### Official Review · Reviewer_zyFn · 2025-11-01

**Soundness:** 2
**Presentation:** 2
**Contribution:** 3
**Rating:** 6
**Confidence:** 2

**Summary:**

This paper studies efficient federated learning under heterogeneous data distributions. The authors decompose data heterogeneity into two structural types — mode-based and coordinate-based — and design theoretically optimal algorithms that decouple communication complexity along these two structural dimensions.

**Strengths:**

1.	The paper provides a novel theoretical perspective by decomposing data heterogeneity into distinct structural forms.

2.	It offers detailed and rigorous theoretical analysis with formal convergence and optimality results.

**Weaknesses:**

1.	The overall organization and presentation are highly theoretical, making the main ideas and algorithmic insights difficult to follow.

2.	The evaluation is limited in scope, using relatively simple models and datasets that may not fully demonstrate the practical value of the proposed approach.

**Questions:**

See the Weaknesses.

---

> ### Author Response · Authors · 2025-11-21
>
> Dear Reviewer zyFn!
>
> Thank you for positive evaluation and for highlighting the strengths of our work! Below, we discuss the weaknesses.
>
> >The overall organization and presentation are highly theoretical, making the main ideas and algorithmic insights difficult to follow.
>
> We agree that our contribution is primarily theoretical. However, the manuscript is extensively supplemented with explanations. For example, the idea of Hessian similarity is explained informally in Lines 58–64; structured heterogeneity is illustrated with examples in Lines 80–84 and Lines 108–111; and the mirror-based step for the data-similarity setup is motivated in Lines 269–276. Algorithmic insights are also motivated informally (see Lines 280–283 for the intuition behind introducing the reference point and how its update frequency reflects the server-side data distribution). Theoretical findings are also explained in detail. For instance, Lines 305–308 discuss why the obtained bound is adequate for the purposes of the paper and why exactly the chosen update of $e^t$ resolves the main theoretical challenge of our work. Algorithm 2 is supported by analogous clarifications.
>
> Thus, we respectfully highlight to Reviewer that the manuscript contains a substantial amount of explanatory material intended to facilitate reading for experts from other subfields of machine learning who may not be familiar with optimization.
>
> >The evaluation is limited in scope, using relatively simple models and datasets that may not fully demonstrate the practical value of the proposed approach.
>
> We agree that extending the experimental section to more complex tasks would strengthen our work. We address the problem of fine-tuning FasterViT 270M for Food101 classification. The table below shows the dynamics of test accuracy.
> |# of $\nabla f$ calls|500|1000|1500|2000|2250|
> |-|-|-|-|-|-|
> |Adam|46.53|62.46|68.72|73.90|74.43|
> |HASCA|47.40|63.74|71.26|74.85|75.40|
>
> We have added this experiment to Appendix A6.
>
> We also ask Reviewer to check the revised PDF, which now includes ablations demonstrating how the theoretical intuition carries over to practice. See Appendix A3 for the practical discussion of the choice of $p$, and Appendix A5 for the analysis of how our approach relates to similarity-accounted competitors. In addition, the revised PDF contains a dedicated section discussing the issue of inexact inner minimization, both in theory and practice (see Appendix H).
>
> We once again thank Reviewer for the positive evaluation of our work. However, we kindly ask them to reconsider their score in light of additional studies we have conducted and the clarifications we have provided regarding the presentation of results.

---

> ### Author Response · Authors · 2025-11-27
>
> Dear Reviewer zyFn! We hope this message reaches you at a convenient time.
>
> We would like to kindly follow up on our rebuttal, as we have not yet received a response from Reviewer. We fully understand that the discussion period requires substantial time and effort. However, given Reviewer's borderline evaluation, their feedback on our reply is particularly important, and we would greatly appreciate the opportunity to initiate the discussion.
>
> Thanks to Reviewer for their time and consideration!

---

### Official Review · Reviewer_4mqV · 2025-11-02

**Soundness:** 3
**Presentation:** 3
**Contribution:** 2
**Rating:** 4
**Confidence:** 4

**Summary:**

This paper addresses the challenge of data heterogeneity in federated learning by proposing an innovative structured solution. The authors identify two heterogeneity patterns: heterogeneity based on data distribution and heterogeneity based on model parameters, and design corresponding HASCA and C-HASCA algorithms. These algorithms achieve effective separation of communication complexity by employing differentiated communication frequencies for different heterogeneous components.

**Strengths:**

1. This paper delves into the inherent structural differences in heterogeneity, namely uneven pattern distribution and varying parameter block characteristics. This deconstruction provides a novel theoretical foundation for designing more refined and efficient algorithms.

2. The core mechanism of HASCA and C-HASCA is a general framework. This idea can be easily transferred to other distributed optimization scenarios beyond federated learning.

**Weaknesses:**

1. The theoretical complexity analysis in this paper primarily focuses on the number of communication rounds or communication attempts. However, in practical federated learning systems, communication bottlenecks often manifest as bandwidth consumption and latency. The paper lacks a quantitative assessment of actual communication time and the total amount of data transmitted.

2. The theoretical analysis is based on the strong assumption that "the server can solve the subproblems precisely" (Algorithm 1, Line 3), but the experimental section mentions that "usually only approximate solutions are needed" (Section 7). The impact of this approximate solution on convergence is not fully discussed in the theoretical section, creating a gap between theory and practice. Furthermore, engineering techniques such as momentum and learning rate scheduling introduced in the algorithm (Section 7.1) are not included in the theoretical framework.

3. The degree of heterogeneity in the experiments (e.g., κ=0.8) is artificially set. How does the algorithm perform in more extreme cases (e.g., some client data is completely unrelated to server data, i.e., δg is maximum)? The robustness analysis in this paper (Appendix A.3) mainly focuses on the proportion of categories covered by the server data, and the analysis of the systematic impact of extreme changes in δ values ​​is insufficient.

**Questions:**

See weeknesses

---

> ### Author Response · Authors · 2025-11-21
>
> Dear Reviewer 4mqV!
>
> Thank you for your time and comments! Further, we are addressing your concerns.
>
> >... analysis in this paper primarily focuses on the number of communication rounds or communication attempts.
>
> Communication rounds is a standard metric for analysis of distributed algorithms in synchronous setup [1, 2]. Since federated networks are typically asynchronous, we also discuss convergence in terms of client–server vector exchanges (Lines 326-330) and number of transmitted bits (Lines 391-396). The description of metrics can be found in Section 2.
>
> >paper lacks a quantitative assessment of actual communication time and the total amount of data transmitted.
>
> We agree that a numerical analysis of runtime is important for demonstrating the practical applicability of our method. We provide results for runs in a federated network of 10 clients with $\kappa = 0.8$. The average device bandwidth is 25Mbps.
> |Time(Sec)|1.25K|2.5K|5K|7.5K|10K|12.5K|15K|
> |-|-|-|-|-|-|-|-|
> |ExtraGrad|30.45|34.42|40.13|43.10|45.47|45.70|46.20|
> |MirrorProx|37.27|47.84|57.79|65.86|69.26|71.17|72.04|
> |Adam|50.54|60.83|66.71|70.61|71.61|73.42|79.54|
> |FedProx|64.66|67.16|73.62|75.54|76.71|76.98|77.71|
> |SCAFFOLD|61.86|65.62|71.82|71.85|72.98|73.91|74.24|
> |HASCA|42.93|55.39|69.43|73.60|77.22|79.02|81.02|
>
> We observe the superiority of the proposed approach. Experiments for federated networks with $\kappa=0.6$ and $\kappa=0.95$ can be found in Appendix A5.
>
> >The impact of this approximate solution on convergence is not fully discussed in the theoretical section, creating a gap between theory and practice
>
> The problem in Line 3 of Algorithm 1 (similarly for Algorithm 2) can be rewritten as $$x^{t+1}=prox_{\theta h_1}(x^t-\theta e^t).$$
> In many top-tier papers, an exact proximal oracle is considered to be available (see Algorithm 1 in [3]; Algorithm 3 in [4]; Algorithms in [5]). Nevertheless, we have modified proofs for both methods to account for inexactness. We derived a precision criterion that must be satisfied to ensure convergence. We also provide an ablation study to empirically demonstrate the robustness of the optimization trajectory to inaccuracies in the inner minimization. We vary the number of epochs to solve the inner problem.
> |Epochs|Final test accuracy|
> |-|-|
> |2|83.69|
> |3|86.82|
> |4|80.5|
>
> See Appendix H for the theory and numerical validation.
>
> > momentum and learning rate scheduling introduced in the algorithm (Section 7.1) are not included in the theoretical framework.
>
> We would like to point out that Algorithms 1,2 have optimal guarantees without the use of aforementioned techniques. Engineering approaches are practical tools aimed at finding the deepest local minimum. In the nonconvex case, it is known from literature that momentum does not provide theoretical improvements. For example, in [6], guarantees for Muon remain identical regardless of whether momentum is considered (Theorems 3.1 and 3.2). Moreover, if it did not smooth the gradient history ($M_t=\nabla f_{S_t}(W_t)$), rates would omit one of irreducible terms (see proof of Theorem 3.1). A similar effect can be observed in [7, 8]. Scheduling may sometimes be crucial for analysis [9, 2], but we also do not need it to obtain desired guarantees.
>
> >The degree of heterogeneity in the experiments (e.g., κ=0.8) is artificially set.
>
> We politely note that, in addition to $\kappa=0.8$, we also investigated homogeneous ($\kappa=0.6$, i.e. $\delta_f\approx\delta_g$) and extremely heterogeneous ($\kappa=0.95$, i.e. $\delta_{f}\ll\delta_g$) cases in the initial submission, see Appendix A2.
>
> We also encourage Reviewer to check the experimental section of the revised PDF for new baselines: FedProx, SCAFFOLD
>
> If Reviewer has any remaining questions, we would be happy to address them during the discussion period. If all aspects have been clarified to Reviewer's satisfaction, we would kindly ask them to reconsider the evaluation.
>
> ---
>
> **References:**
>
> [1] Richtarik, P., et al. “EF21: A New, Simpler, Theoretically Better, and Practically Faster Error Feedback”. **NeurIPS-21**.
>
> [2] Kovalev, D., et al. “Optimal Gradient Sliding and its Application to Distributed Optimization Under Similarity”. **NeurIPS-22**.
>
> [3] Mischenko, K., et al. “ProxSkip: Yes! Local Gradient Steps Provably Lead
> to Communication Acceleration! Finally!”. **ICML-22**
>
> [4] Malitsky, Y., Mischenko, K. “Adaptive Proximal Gradient Method for Convex
> Optimization”. **NeurIPS-24**
>
> [5] Traore, C., et al. “ Variance Reduction Techniques for Stochastic Proximal Point Algorithms”. **JOTA**, 2024
>
> [6] Sato, N., et al. “Analysis of Muon’s Convergence and Critical Batch Size”. Preprint, July 2025
>
> [7] Shen, W., et al. “On the Convergence Analysis of Muon”. Preprint, May 2025
>
> [8] Xie, S., et al. “From PowerSGD to PowerSGD+: Low-Rank Gradient Compression for Distributed Optimization with Convergence Guarantees”. Preprint, September 2025
>
> [9] Gower, R., et al. “SGD: General Analysis and Improved Rates”. **ICML-19**

---

> ### Author Response · Authors · 2025-11-27
>
> Dear Reviewer 4mqV! We hope this message finds you well.
>
> We would like to kindly follow up regarding our rebuttal, as we have not yet received a response. We understand that author-reviewer discussion is demanding, and we sincerely appreciate the time and effort Reviewer dedicate to evaluating the submissions. If Reviewer has an opportunity, we would be glad to receive a feedback on our rebuttal and initiate a discussion.
>
> Thanks to Reviewer for their time and consideration!

---

### Meta-Review · Area_Chair_oo3n · 2026-01-05

**Summary:**

The paper decomposes data heterogeneity in federated learning into mode‑based and coordinate‑based structures and introduces algorithms that exploit this structure to reduce communication frequency while preserving convergence. Most reviewers appreciate the authors’ effort to delve into the inherent structural differences in heterogeneity and to provide detailed theoretical analysis with formal convergence and optimality results. Several common concerns were raised and partially addressed, including the limited scope of evaluation (in terms of both models and datasets) and the strong assumptions required for the analysis.

**Reviewer Concerns:**

Multiple common concerns were raised by the reviewers, including the limited scope of evaluation (in terms of both models and datasets), several strong assumptions required for the analysis, and the lack of a quantitative assessment of actual communication time and total data transmitted. During the rebuttal, the authors provided additional runtime results, added an experiment on Food101 classification, and clarified the notation confusion between O(1/\epsilon ^2) and O(1/\epsilon ), which partially addressed these concerns.

**Reviewer Scores:**

This paper receives the following ratings: Marginally Below, Marginally Above, Strong Reject, and Marginally Above. If the reviewers had been able to participate fully in the discussion, the AC would expect some negative ratings likely to remain. The AC recommends not accepting the paper.

---

### Decision · Program_Chairs · 2026-01-26

Reject